# Epitope-associated and specificity-focused features of EV71-neutralizing antibody repertoires from plasmablasts of infected children

Kuan-Ying Arthur Huang [1], Mei-Feng Chen[2,3], Yhu-Chering Huang[1], Shin-Ru Shih[2], Cheng-Hsun Chiu[1,4], Jainn-Jim Lin[5], Jen-Ren Wang[6], Kuo-Chien Tsao[7] & Tzou-Yien Lin[1,8]

Protective antibody levels are critical for protection from severe enterovirus 71 infection. However, little is known about the specificities and functional properties of the enterovirus 71-specific antibodies induced by natural infection in humans. Here we characterize 191 plasmablast-derived monoclonal antibodies from three enterovirus 71-infected children, each of whom shows a distinct serological response. Of the 84 enterovirus 71-specific antibodies, neutralizing antibodies that target the rims and floor of the capsid canyon exhibit broad and potent activities at the nanogram level against viruses isolated in 1998–2016. We also find a subset of infected children whose enterovirus 71-specific antibodies are focused on the 3- and 2-fold plateau epitopes localized at the margin of pentamers, and this type of antibody response is associated with lower serum titers against recently circulating strains. Our data provide new insights into the enterovirus 71-specific antibodies induced by natural infection at the serological and clonal levels.

[1] Division of Infectious Diseases, Department of Pediatrics, Chang Gung Memorial Hospital, Taoyuan 33305, Taiwan. [2] Research Center for Emerging Viral Infections, Chang Gung University, Taoyuan 33302, Taiwan. [3] Bone and Joint Research Center, Chang Gung Memorial Hospital, Taoyuan 33305, Taiwan. [4] Molecular Infectious Disease Research Centre, Chang Gung Memorial Hospital, Taoyuan 33305, Taiwan. [5] Division of Pediatric Critical Care, Department of Pediatrics, Chang Gung Memorial Hospital, Taoyuan 33305, Taiwan. [6] Department of Medical Laboratory Science and Biotechnology, College of Medicine, National Cheng Kung University, Tainan 70101, Taiwan. [7] Department of Laboratory Medicine, Chang Gung Memorial Hospital, Taoyuan 33305, Taiwan. [8] College of Medicine, Chang Gung University, Taoyuan 33302, Taiwan. Correspondence and requests for materials should be addressed to K.-Y.A.H. (email: arthur1726@cgmh.org.tw) or to T.-Y.L. (email: pidlin@cgmh.org.tw)

Enterovirus 71 (EV71), a non-enveloped single-strand RNA virus, is a leading cause of hand-foot-and-mouth disease and herpangina in children in Europe and the Asia–Pacific region. Outbreaks of EV71 infection are also associated with meningitis, polio-like syndrome, encephalitis with subsequent cardiopulmonary collapse, and mortality[1–5]. In 2011, an outbreak occurred in Vietnam, and 174,677 cases and 200 deaths were reported[2]. In another outbreak in China in 2012, more than two million cases and hundreds of deaths were reported[4]. According to the European Centre for Disease Prevention and Control, an outbreak in Catalonia, Spain in 2016 resulted in 109 severe cases. Despite the severity of EV71 infection, there are no licensed vaccines in most countries affected by EV71 and no specific EV71 therapeutics.

Typically, EV71 outbreaks occur in the Asia–Pacific region every 2–3 years, and the emergence of new strains via genotype replacement or switching makes EV71 a constant threat to children's health[6–9]. High rates of mutation and genomic recombination are common characteristics of enteroviruses, and inter- and intra-typic recombination events are frequently detected in the EV71 strains responsible for outbreaks[7–12].

The emergence of novel EV71 strains and genotypes could be due to viral evasion of host immunity. This notion is firstly supported by longitudinal and population-scale serological studies showing that a lack of protective antibody levels was responsible for the high infection rate and increased risk of severe illness in young children in the genotype C2 EV71 outbreak in Taiwan in 1998[13]. Similar interpretations were made in the pre- and post-outbreak serum surveys for the emergence of genotype B5 EV71 in Thailand in 2012[14]. Secondly, in some children, acute EV71 infection elicits only a low neutralization titer or a genotype-biased antibody response, although cross-neutralization has also been detected in human sera[6, 15–18]. Several models have shown that antibody-mediated neutralization plays a key role in the selection of viral variants[6, 8, 19–21]. Consistent with this, a recent study using panels of murine EV71-specific monoclonal antibodies (mAbs) demonstrated considerable antigenic diversity among EV71 isolates and genotypes[22]. However, the antibody specificities that contribute to the serum-neutralizing activity against EV71 are unclear. This issue is particularly relevant for outbreak control and vaccine design. However, increasing evidence has shown that EV71 vaccines may not provide cross-protection against all circulating genetic lineages of virus[17, 23].

The EV71 capsid is assembled from 60 subunits of four proteins (VP1–4); VP1, VP2, and VP3 are exposed on the outside of the capsid, while VP4 is internal[24, 25]. Linear neutralizing epitopes, consisting of residues 215–220 of the VP1 GH loop, residues 141–146 of the VP2 EF loop, and the conformational epitope in the VP3 knob region, have been characterized in previous studies of murine cross-neutralizing mAbs[26–31]. Residue 145 on the surface of VP1 has been identified as a key antigenic determinant of strain-specific murine mAbs[32]. Nevertheless, such mAbs do not necessarily reflect the naturally acquired EV71-specific antibody repertoire in children. To date, no human EV71-specific mAbs have been reported; thus, the neutralizing epitopes targeted by the human antibody response during acute EV71 infection are undetermined.

During acute EV71 infection in pediatric patients, robust induction of virus-specific IgG plasmablasts is detected in the first week, and the response peaks on 4–7 days after symptom onset[15]. While cross-neutralization between genotype B and C viruses has been observed, some post-infection sera from children infected with genotype B viruses show substantially reduced titers against genotype C EV71[15]. In this study, we characterized a large panel of plasmablast-derived IgG mAbs that targeted the capsid of EV71 to unravel the relevant epitopes for the neutralizing

antibodies induced by natural infection. Potent and broadly reactive antibodies recognized novel neutralizing epitopes on the floor and rims of the capsid canyon. However, antibodies that recognized the 3- and 2-fold plateau epitopes on the margin of pentamers had limited neutralization breadth and 10–100-fold lower potency. We also demonstrated that a predominance of suboptimal antibody clones that failed to neutralize or only weakly neutralized EV71, including the latest circulating genotype, was associated with the low serological titer observed in a subset of children.

## Results

**The EV71-neutralizing antibody response to natural infection.** Three hospitalized children, referred to as donors M, Y, and Z, had hand-foot-and-mouth disease with laboratory-confirmed genotype B5 EV71 infection in 2012 (Supplementary Fig. 1). The ages of donors M, Y, and Z were 4, 6, and 3 years, respectively. All three donors had febrile illnesses for 4 days, and none developed neurological manifestations. The donors generated comparable EV71-specific IgG plasmablast responses on day 7 after illness onset (average frequency ± standard deviation of two replicates, 830 ± 28 per million peripheral blood mononuclear cells for donor M, 620 ± 28 for donor Y, and 715 ± 21 for donor Z; $P = 0.1017$, ANOVA; Supplementary Fig. 2).

Paired sera showed a 4- to 32-fold increase in neutralization titer to the 2012 EV71 strain 12-96015 in three donors (Fig. 1a). In 2012, coxsackievirus A2 co-circulated with EV71 in Taiwan,

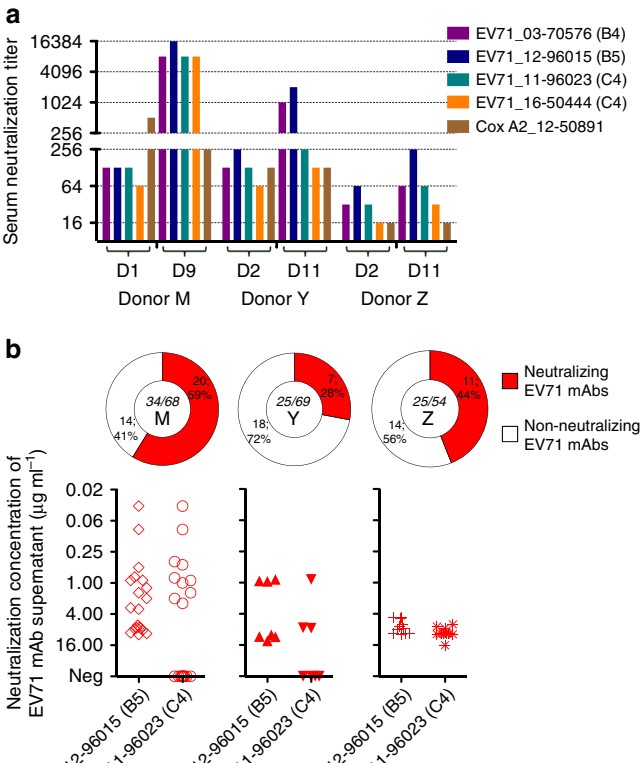

**Fig. 1** Serological neutralization titers for EV71 and virus-specific plasmablast-derived mAbs from three children with laboratory-confirmed genotype B5 EV71 infection. **a** Paired sera were collected from donors and tested against a selection of clinical EV71 strains. **b** IgG plasmablasts collected on day 7 after illness onset were used to produce mAbs, and 34/68 antibodies from donor M, 25/69 antibodies from donor Y, and 25/54 antibodies from donor Z were specific to EV71. The neutralization assay was carried out twice with equivalent results

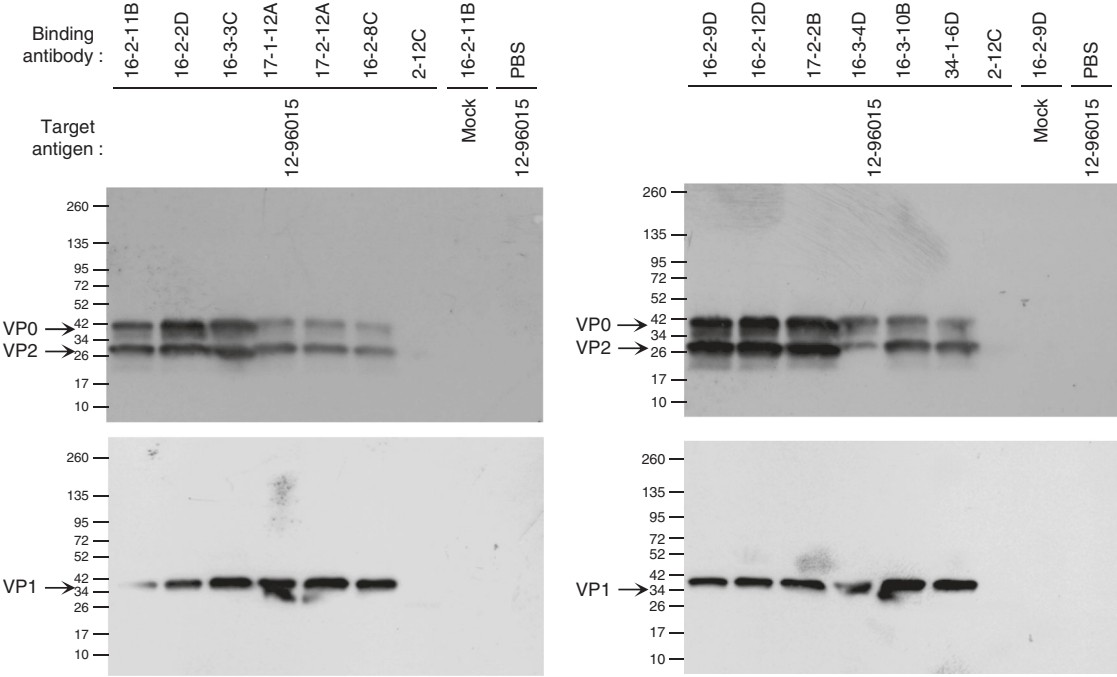

**Fig. 2** Characterization of EV71-neutralizing mAbs. Immunoprecipitated EV71 capsids were detected by immunoblot analysis using the anti-VP0/VP2 monoclonal antibody MAB979 (1:1000 dilution, EMD Millipore) and the anti-VP1 monoclonal antibody MAB1255-M05 (1:1000 dilution, Abnova). The 12-96015 EV71-containing and mock-infected RD cell supernatants were prepared as antigens and immunoprecipitated with the anti-EV71 mAbs. The human anti-influenza monoclonal antibody 2-12C[19] was included as an antibody control. For the mock control, all antibodies were set up and showed equivalent results. In the figure, one mock control is shown

and no titer change against coxsackievirus A2 was detected in paired sera.

We further tested the paired sera against a selection of EV71 viruses, including a 2003 genotype B4 strain, 2011 C4 strain, and another recently circulating C4 strain. Each donor developed a distinct serological antibody response pattern, in terms of magnitude and breadth, at 9–11 days after onset. Figure 1a shows strong, broad cross-neutralizing serological responses against genotypes B and C in donor M, preferential responses against genotype B viruses in donor Y, and a small increase in neutralization titers to EV71 in donor Z. These serological data suggest that the antibodies produced in response to infection in donors Y and Z neutralize B5 virus but may fail to neutralize the 2011 and 2016 C4 viruses.

To explore the clonal basis of the discrepancies in the titer and breadth of infection-induced antibody responses among the donors, we identified the circulating plasmablasts present on day 7 after onset by flow cytometry and sorted single cells to generate human IgG mAbs.

**Plasmablast-derived EV71-specific mAbs**. A total of 191 antibodies were produced by the three donors, of these 34 out of 68 (50%) from donor M, 25 out of 69 (36%) from donor Y, and 25 out of 54 (46%) antibodies from donor Z ($P = 0.2467$, $\chi^2$ test) recognized EV71, as shown by staining of virus-infected cells by flow cytometry and binding of purified viruses by ELISA (Fig. 1b). These EV71-specific antibodies were able to immunoprecipitate viral capsids (Fig. 2 and Supplementary Fig. 3) but did not recognize denatured proteins on an immunoblot under reducing or non-reducing conditions. These results suggest that the EV71-specific antibodies elicited upon natural infection recognize the complex conformational epitopes of the viral capsid.

Of the 84 EV71-specific antibodies, 38 neutralized EV71 viruses (Fig. 1b and Table 1). There was no significant difference

in the percentage of neutralizing antibody clones among donors (59% of EV71-specific IgG plasmablasts in donor M, 28% in donor Y, and 44% in donor Z; $P = 0.0624$, $\chi^2$ test). However, while a large proportion of the neutralizing clones from donor M (10 out of 20, 50%) showed high-potency cross-reactivity with genotype B5 and C4 viruses, most neutralizing clones from donors Y (4 out of 7, 57%) and Z (11 out of 11, 100%) showed substantially lower potencies (Fig. 1b and Table 1 column 12), which corresponded to poor serological responses to EV71 viruses, particularly genotype C4 viruses, in these two donors (Fig. 1a).

The variable domain sequences were obtained from the 84 EV71-specific antibodies, and each was unique and harbored somatic mutations (Table 1 and Supplementary Table 1). No significant differences in the numbers of nucleotide mutations and amino acid replacements in the heavy-chain variable domains were found between neutralizing and non-neutralizing antibodies (Supplementary Fig. 4). The rapid acquisition of antigen specificity and accumulation of somatic mutations indicated that these plasmablasts were probably differentiated from pre-existing memory B cells.

All three EV71-specific antibody repertoires were highly diverse, containing 12–21 unique combinatorial VDJ rearrangements in the heavy-chain variable domain (Table 1 and Supplementary Table 1). Compared to non-neutralizing antibodies, the neutralizing antibody repertoires were skewed toward six germline variable region gene segments ($V_H$ 1-46, 3-9, 3-11, 3-23, 4-39, and 7-4-1). We did not observe a prominent family of convergent gene usage in the neutralizing antibodies from donors M and Y, whereas a bias toward $V_H$ 7-4-1*02/$D_H$ 1-7*01/$J_H$ 6*02 and $V_{\hat{k}}$ 1-39*01/$J_{\hat{k}}$ 4*01 was found in the neutralizing antibodies from donor Z. Biased gene usage in the variable domain has also been observed in influenza hemagglutinin (HA) stem-, HIV V3-, and rotavirus-specific antibodies derived from infected donors[33–35].

Table 1 shows that the 38 neutralizing antibodies were evolved from 12 clonal groups, as defined by their VDJ and VJ rearrangements. To further investigate the functional activities of these antibodies and map the relevant epitopes, representative mAbs from each of 12 variable domain-related groups were expanded, purified, and characterized in detail. These 12

### Table 1 Neutralizing EV71 capsid-specific human monoclonal antibodies

| mAb[a] | $V_H$ | $D_H$ | $J_H$ | Heavy-chain junction sequence | Mut[b] | λ/κ | $V_κ/V_λ$ | $J_κ/J_λ$ | Light-chain junction sequence | Mut[b] | Neut (μg ml$^{-1}$)[c] |
|---|---|---|---|---|---|---|---|---|---|---|---|
| *Donor M* | | | | | | | | | | | |
| *Group 1* | | | | | | | | | | | |
| **16-2-11B** | 1-46*01 or 03 | 2-21*02 | 4*02 | CARNYNGYCAGDCYSPDFW | 23 (16) | λ | 2-14*01 | 2*01 or 3*01 or 3*02 | CSAFTTSSTLVF | 18 (12) | 0.78/0.39 |
| 16-1-10B | 1-46*01 or 03 | 2-21*02 | 4*02 | CARNYNGYCAGECYSPDYW | 23 (10) | λ | 2-14*01 | 2*01 or 3*01 or 3*02 | CSSFTTSSTLVF | 20 (14) | 0.89/0.45 |
| *Group 2* | | | | | | | | | | | |
| **16-2-2D** | 1-46*01 or 02 or 03 | 3-22*01 | 6*02 | CARGPGPGGKYYYDSSDAYYYYGMDVW | 28 (18) | λ | 1-44*01 | 1*01 | CAAWDDRLNAYVF | 13 (7) | 0.50/2.01 |
| *Group 3* | | | | | | | | | | | |
| **16-3-3C** | 3-9*01 | 6-19*01 | 6*02 | CAKDGPSSGWSYQNYYNAMDVW | 22 (14) | λ | 2-11*01 | 2*01 or 3*01 or 3*02 | CCSYAGSDTLVF | 12 (8) | 2.01/1.00 |
| 16-3-5C | 3-9*01 | 6-19*01 | 6*02 | CAKDGPSSGWSYQNYYNAMDVW | 18 (11) | λ | 2-11*01 | 2*01 or 3*01 or 3*02 | CCSYAGSDTLVF | 9 (7) | 1.25/2.50 |
| *Group 4* | | | | | | | | | | | |
| **16-2-12D** | 4-39*01 | 2-2*01 | 4*02 | CARHASPHCSSTSCYDGPYNKNWYVDLW | 21 (15) | λ | 1-47*02 | 2*01 or 3*01 | CAAWDDSLSGLVF | 13 (4) | 1.55/1.55 |
| *Group 5* | | | | | | | | | | | |
| **16-2-9D** | 4-39*01 or 02 | 2-2*01 | 4*02 | CARHSSPQCSPTSCYEGPYTRDWYVDYW | 24 (16) | λ | 1-44*01 | 2*01 or 3*01 | CAAWDGSLNAVVF | 13 (8) | 0.90/0.90 |
| 16-2-6B | 4-39*01 or 02 | 2-2*01 | 4*02 | CARHSSPQCSPTSCYEGPYTRNWYVDYW | 23 (17) | λ | 1-44*01 | 2*01 or 3*01 | CAAWDDSLNAVVF | 8 (4) | 3.22/0.81 |
| *Group 6* | | | | | | | | | | | |
| 16-1-4A | 4-39*01 | 6-19*01 | 6*02 | CARHVPVAGFGYYYYGMDVW | 19 (11) | λ | 1-44*01 | 3*02 | CAAWDDSLNNWVF | 10 (8) | 8.25/− |
| 16-1-7A | 4-39*01 | 6-19*01 | 6*02 | CARHVPVAGFGYYYYGMDVW | 22 (12) | λ | 1-44*01 | 3*02 | CAAWDDSLNNWVF | 11 (8) | 6.60/− |
| 16-1-8A | 4-39*01 | 6-19*01 | 6*02 | CARHVPVAGFGYYYYGMDVW | 23 (12) | λ | 1-44*01 | 3*02 | CAAWDDSLNNWVF | 9 (7) | 10.00/− |
| 16-1-3B | 4-39*01 | 6-19*01 | 6*02 | CARHVPVAGFGYYYYGMDVW | 25 (13) | λ | 1-44*01 | 3*02 | CAAWDDSLNNWVF | 9 (7) | 7.15/− |
| 16-1-12B | 4-39*01 | 6-19*01 | 6*02 | CARHVPVAGFGYYYYGMDVW | 24 (13) | λ | 1-44*01 | 3*02 | CAAWDDSLNNWVF | 11 (8) | 9.25/− |
| 16-2-10A | 4-39*01 | 6-19*01 | 6*02 | CARHVPVAGFGYYYYGMDVW | 25 (14) | λ | 1-44*01 | 3*02 | CAAWDDSLNNWVF | 10 (7) | 9.55/− |
| 16-2-11D | 4-39*01 | 6-19*01 | 6*02 | CARHVPVAGFGYYYYGMDVW | 23 (15) | λ | 1-44*01 | 3*02 | CAAWDDSLNTWVF | 10 (7) | 7.40/− |
| 16-3-10A | 4-39*01 | 6-19*01 | 6*02 | CARHVPVAGFGYYYYGMDVW | 22 (13) | λ | 1-44*01 | 3*02 | CAAWDDSLNNWVF | 12 (8) | 7.60/− |
| **16-3-4D** | 4-39*01 | 6-19*01 | 6*02 | CARHVPVAGFGYYYYGMDVW | 21 (12) | λ | 1-44*01 | 3*02 | CAAWDDSLNNWVF | 10 (7) | 3.05/− |
| 16-3-8D | 4-39*01 | 6-19*01 | 6*02 | CARHVPVAGFGYYYYGMDVW | 27 (15) | λ | 1-44*01 | 3*02 | CAAWDDSLNNWVF | 10 (7) | 7.75/− |
| *Group 7* | | | | | | | | | | | |
| **16-2-8C** | 4-39*03 | 2-2*01 | 4*02 | CVRHSSPQCSPTSCYEGPYTRDWYVDYW | 28 (16) | λ | 1-44*01 | 2*01 or 3*01 | CAAWDGSLNAVVF | 13 (8) | 0.09/0.09 |
| *Group 8* | | | | | | | | | | | |
| **16-3-10B** | 7-4-1*02 | 1-14*01 | 5*02 | CAYDPLGNWFDPW | 21 (12) | λ | 2-23*01 or 03 | 1*01 | CCSYAGTRTYVF | 16 (9) | 0.03/0.03 |
| *Donor Y* | | | | | | | | | | | |
| *Group 1* | | | | | | | | | | | |
| **17-1-12A** | 3-11*05 | 1-26*01 | 6*02 | CAREKWEKLGKLYYYGLDVW | 30 (20) | κ | 2-28*01 or 2D-28*01 | 2*02 | CMQALQTPRTF | 6 (4) | 0.93/7.45 |
| 17-3-5A | 3-11*05 | 1-26*01 | 6*02 | CAREKWEKLGKLYYYGLDVW | 25 (18) | κ | 2-28*01 or 2D-28*01 | 2*02 | CMQALQTPRTF | 7 (4) | 0.95/7.60 |
| *Group 2* | | | | | | | | | | | |
| **17-2-12A** | 3-23*04 | 3-16*01 | 6*02 | CAKSVAARRFYFYYGMDAW | 28 (18) | λ | 7-43*01 | 3*02 | CLLYYGGSQLWVF | 14 (8) | 10.25/− |
| 17-3-2A | 3-23*04 | 3-16*01 | 6*02 | CAKSVAARRFYFYYGMDAW | 22 (17) | λ | 7-43*01 | 3*02 | CLLYYGGSQLWVF | 13 (8) | 11.10/− |
| 17-3-10A | 3-23*04 | 3-16*01 | 6*02 | CAKSVAARRFYFYYGMDAW | 27 (16) | λ | 7-43*01 | 3*02 | CLLYYGGSQLWVF | 12 (6) | 13.40/− |
| 17-3-11D | 3-23*04 | 3-16*01 | 6*02 | CAKSVAARRFYFYYGMDAW | 24 (16) | λ | 7-43*01 | 3*02 | CLLYYGGSQLWVF | 9 (7) | 11.15/− |
| *Group 3* | | | | | | | | | | | |
| **17-2-2B** | 4-39*01 | 3-10*01 | 4*02 | CARTYGSGSYWGYFEYW | 3 (3) | λ | 2-8*01 | 3*02 | CSSYAGSNNWVF | 1 (0) | 0.87/0.87 |
| *Donor Z* | | | | | | | | | | | |
| *Group 1* | | | | | | | | | | | |
| **34-1-6D** | 7-4-1*02 | 1-7*01 | 6*02 | CARAKALLYYGMDVW | 4 (3) | κ | 1-39*01 or 1D-39*01 | 4*01 | CQQSYSTPLTF | 0 (0) | 4.70/9.40 |
| 34-2-3A | 7-4-1*02 | 1-7*01 | 6*02 | CARAKALLYYGLDVW | 3 (2) | κ | 1-39*01 or 1D-39*01 | 4*01 | CQQSYSTPLTF | 1 (0) | 9.75/9.75 |
| 34-2-5A | 7-4-1*02 | 1-7*01 | 6*02 | CARAKALLYYGMDVW | 5 (3) | κ | 1-39*01 or 1D-39*01 | 4*01 | CQQSYSSPLTF | 3 (1) | 4.95/9.90 |
| 34-2-9B | 7-4-1*02 | 1-7*01 | 6*02 | CARAKALLYYGMDVW | 2 (1) | κ | 1-39*01 or 1D-39*01 | 4*01 | CQQSYSSPLTF | 2 (1) | 4.78/9.55 |
| 34-2-1C | 7-4-1*02 | 1-7*01 | 6*02 | CARAKALLYYGLDVW | 2 (2) | κ | 1-39*01 or 1D-39*01 | 4*01 | CQQSYSTPLTF | 2 (0) | 7.10/7.10 |
| 34-2-5D | 7-4-1*02 | 1-7*01 | 6*02 | CARAKALLYYGMDVW | 6 (3) | κ | 1-39*01 or 1D-39*01 | 4*01 | CQQSYSSPLTF | 1 (0) | 6.45/6.45 |
| 34-3-4A | 7-4-1*02 | 1-7*01 | 6*02 | CARAKALLYYGMDVW | 1 (1) | κ | 1-39*01 or 1D-39*01 | 4*01 | CQQSYSSPLTF | 3 (1) | 9.95/9.95 |
| 34-3-4B | 7-4-1*02 | 1-7*01 | 6*02 | CARAKALLYYGLDVW | 7 (4) | κ | 1-39*01 or 1D-39*01 | 4*01 | CQQSYSSPLTF | 0 (0) | 9.70/9.70 |
| 34-3-6B | 7-4-1*02 | 1-7*01 | 6*02 | CARAKALLYYGLDVW | 5 (4) | κ | 1-39*01 or 1D-39*01 | 4*01 | CQQSYSSPLTF | 0 (0) | 9.65/9.65 |
| 34-3-1C | 7-4-1*02 | 1-7*01 | 6*02 | CARAKALLYYGMDVW | 4 (3) | κ | 1-39*01 or 1D-39*01 | 4*01 | CQQSYSTPLTF | 1 (0) | 8.10/16.20 |
| 34-3-8D | 7-4-1*02 | 1-7*01 | 6*02 | CARAKALLYYGMDVW | 1 (1) | κ | 1-39*01 or 1D-39*01 | 4*01 | CQQSYSTPLTF | 2 (1) | 7.65/7.65 |

$D_H$, diversity gene segment of the heavy-chain variable domain; $J_H$, joining gene segment of the heavy-chain variable domain; Mut, mutation number; $J_λ$ joining gene segment of the lambda light-chain variable domain; Neut, neutralization; $V_H$, variable gene segment of the heavy-chain variable domain; $V_κ$, variable gene segment of the kappa light-chain variable domain; $V_λ$, variable gene segment of the lambda light-chain variable domain; $J_κ$ joining gene segment of the kappa light-chain variable domain

[a]Representative neutralizing monoclonal antibodies (mAb) from each of the 12 variable domain-related groups are shown in bold

[b]The number of nucleotide mutations in the heavy- and light-chain variable domains and the number of amino acid replacements (shown in parentheses). The variable domain consists of the framework regions (FR1, FR2, FR3, and FR4) and complementarity determining regions (CDR1, CDR2, and CDR3). To determine the individual gene segments employed by VDJ and VJ rearrangements and the number of nucleotide mutations and amino acid replacements, the variable domain sequences were aligned with germline gene segments using the international ImMunoGeneTics (IMGT) alignment tool

[c]The lowest concentration of the mAb-containing supernatant that showed 100% inhibition of EV71 12-96015 (B5)/11-96023 (C4)-induced CPE. -: no neutralization

**Table 2 Neutralization of EV71 clinical strains in 1998–2016**

| Year | 1998 | 1998 | 1999 | 1999 | 2000 | 2001 | 2002 | 2003 | 2004 | 2005 | 2007 | 2008 | 2010 | 2011 | 2012 | 2014 | 2015 | 2016 | 2016 |
|---|---|---|---|---|---|---|---|---|---|---|---|---|---|---|---|---|---|---|---|
| Strain name | 98-2086 | 98-4215 | 99-1691 | 99-3351 | 00-2278 | 01-1437 | 02-2792 | 03-70576 | 04-72232 | 05-1956 | 07-72043 | 08-96016 | 10-96018 | 11-96023 | 12-96015 | 14-51389 | 15-921 | 16-50444 | 16-50555 |
| Genotype | C2 | C1 | C2 | B4 | B4 | B4 | B4 | B4 | C4 | C4 | C5 | B5 | C4 | C4 | B5 | B5 | B5 | C4 | C4 |
| *Representative mAbs*[a] | | | | | | | | | | | | | | | | | | | |
| 16-3-10B | – | + | – | +++ | ++ | ++ | ++ | +++ | +++ | ++ | ++ | ++++ | ++++ | ++++ | ++++ | +++ | ++++ | +++ | +++ |
| 16-2-8C | – | – | – | ++ | +++ | ++ | +++ | ++ | ++ | ++ | +++ | ++++ | ++++ | ++++ | ++++ | +++ | ++++ | +++ | +++ |
| 16-2-11B | – | + | – | ++ | ++ | ++ | ++ | ++ | +++ | ++ | ++ | +++ | +++ | +++ | +++ | ++ | ++ | ++ | ++ |
| 17-2-2B | – | + | – | ++ | ++ | ++ | ++ | ++ | ++ | ++ | ++ | ++ | ++ | +++ | +++ | +++ | +++ | +++ | +++ |
| 16-2-9D | – | – | – | ++ | ++ | ++ | ++ | ++ | +++ | +++ | ++ | ++ | ++ | +++ | +++ | ++ | ++ | ++ | +++ |
| 16-2-12D | – | – | – | ++ | ++ | +++ | +++ | ++ | + | ++ | ++ | ++ | +++ | ++ | ++ | ++ | ++ | ++ | ++ |
| 16-3-3C | – | + | – | – | – | – | – | ++ | ++ | ++ | ++ | ++ | ++ | ++ | ++ | ++ | ++ | ++ | ++ |
| 16-2-2D | – | – | – | ++ | ++ | ++ | +++ | ++ | + | ++ | + | ++ | ++ | ++ | +++ | +++ | ++ | ++ | ++ |
| 17-1-12A | – | – | – | ++ | ++ | ++ | ++ | ++ | + | ++ | + | ++ | ++ | ++ | +++ | +++ | ++ | ++ | +++ |
| 34-1-6D | – | – | – | ++ | ++ | ++ | ++ | + | + | + | + | ++ | + | ++ | ++ | + | + | + | + |
| 16-3-4D | – | – | – | + | + | + | + | + | – | – | – | + | – | – | ++ | ++ | + | – | – |
| 17-2-12A | – | – | – | + | + | + | + | + | – | – | – | + | – | – | + | + | + | – | – |
| *Controls*[b] | | | | | | | | | | | | | | | | | | | |
| 17-1-10B | – | – | – | – | – | – | – | – | – | – | – | – | – | – | – | – | – | – | – |
| 16-2-1A | – | – | – | – | – | – | – | – | – | – | – | – | – | – | – | – | – | – | – |
| D9 Serum | 1:16 | 1:32 | 1:16 | 1:512 | 1:512 | 1:1024 | 1:512 | 1:512 | 1:256 | 1:512 | 1:512 | 1:2048 | 1:1024 | 1:4096 | 1:4096 | 1:2048 | 1:2048 | 1:1024 | 1:1024 |

[a]concentrations at which 100% neutralization was achieved: ++++, <100 ng ml$^{-1}$; +++, 0.1–1 μg ml$^{-1}$; ++, 1–10 μg ml$^{-1}$; +, 10–50 μg ml$^{-1}$; –, no neutralization

[b]Non-neutralizing EV71 mAbs, 17-1-10B and 16-2-1A, and post-infection serum collected at day 9 after illness onset from a hospitalized child with laboratory-confirmed genotype B5 EV71 infection were included in the test. The neutralizing activity of each representative mAb is defined as the lowest antibody concentration that completely inhibited CPE formation. Each sample dilution was assayed in triplicate, and the assay was carried out twice with equivalent results

representative antibodies showed comparable binding activities to infected cells in the flow cytometry-based assay (Supplementary Fig. 5 and Supplementary Table 2).

**The potency and breadth of neutralizing mAbs.** To explore the neutralization breadth of representative mAbs, they were tested against EV71 clinical strains isolated in 1998–2016. Table 2 shows that 10 of the 12 representative mAbs were broadly reactive against genotypes B (B4 and B5) and C (C1, C4, and C5) and exhibited varying degrees of heterologous neutralization. However, genotype C2 viruses were resistant to the tested representative antibodies, even at 50 μg ml$^{-1}$.

Six representative antibodies (16-3-10B, 16-2-8C, 16-2-11B, 17-2-2B, 16-2-9D, and 16-2-12D) potently neutralized B4, B5, and C4 viruses at <1 μg ml$^{-1}$, and two antibodies (16-3-10B and 16-2-8C) displayed ultra-potent neutralization at <100 ng ml$^{-1}$. However, in most instances, high concentrations (>10 μg ml$^{-1}$) of the cross-reactive antibody 34-1-6D (V$_H$ 7-4-1*02/D$_H$ 1-7*01/J$_H$ 6*02 clonal group) from donor Z were required to neutralize viruses. We also noted that two representative antibodies, 16-3-4D and 17-2-12A, neutralized only genotype B viruses, and in most cases, at concentrations >10 μg ml$^{-1}$.

**Epitopes recognized by neutralizing mAbs.** To identify the critical determinants engaged by the neutralizing mAbs, we selected escape mutants of the 12-96015 (genotype B5) and 11-96023 (genotype C4) viruses in the presence of antibody. Figure 3a shows that 33 of the 35 mAb-escape mutants had a single amino acid mutation that effectively abolished antibody binding and neutralization (Supplementary Fig. 6), while the remaining two mutants had more than one mutation. These mutants showed rhabdomyosarcoma (RD) cell infectivity similar to that of the parent strain.

The mutant analysis results revealed 25 amino acid substitutions at 19 capsid surface residues, and 17 of these residues were conserved among the EV71 genotypes, except VP1 residue 164 (aspartate to glutamate; Fig. 3a) and VP1 residue 283 (serine to threonine in 2016 genotype C4 viruses; Supplementary Fig. 7). The substitution at VP1 residue 164 in genotype B4 viruses is associated with resistance to neutralization by antibody 16-3-3C (Fig. 3a and Table 2). Although VP1 residue 283, together with residue 141, are antigenically relevant to 16-3-10B antibody binding, the antibody could tolerate the amino acid substitution

in 2016 genotype C4 viruses, suggesting that binding to VP1 residue 283 is required but not sufficient for neutralization (Fig. 3a, Supplementary Fig. 7, and Table 2).

One of escape mutants selected from 12-96015 (genotype B5) using antibody 17-2-12A harbored two amino acid substitutions (VP3 residues 144 and 148; Fig. 3a), and these two residues are close to each other in the structure (Fig. 3b). Antibody 17-2-12A appeared to be sensitive to the VP3 K144E change (Fig. 3a and Supplementary Fig. 6). To determine the role of the T148A substitution in antibody binding and neutralization, these two mutations were introduced into an infectious cDNA clone of genotype B5 EV71[36] via site-directed mutagenesis. The VP3 K144E and VP3 T148A substitutions, both together and individually, abolished neutralization and binding by antibody 17-2-12A (Supplementary Fig. 8).

We mapped the 19 surface residues critical for antibody neutralization and showed that they were located in five structural regions of the capsid, designated as the canyon northern rim, canyon floor, canyon southern rim, 3-fold plateau, and 2-fold plateau (Fig. 3b). Neutralizing antibodies were categorized into five subgroups according to their cross-reactivity with the full panel of escape mutants: canyon northern rim (16-2-11B, 16-2-2D, and 16-3-3C), canyon floor (16-2-8C, 16-2-9D, and 16-2-12D), canyon southern rim (17-2-2B and 16-3-10B), 3-fold plateau (16-3-4D and 34-1-6D) and 2-fold plateau (17-1-12A and 17-2-12A; Table 3).

**Canyon- and plateau-specific neutralizing antibodies.** Table 2 and Fig. 4a show that canyon-specific antibodies had broad and potent neutralizing activities against EV71, whereas 3- and 2-fold plateau-specific antibodies had significantly weaker potency and narrower neutralization breadth. The sequence data revealed that canyon-specific antibodies comprised eight clonal groups, which did not show convergent variable domain gene usage or somatic mutation levels (Table 1). Nevertheless, the average heavy-chain CDR3 length of canyon-specific antibodies was significantly longer than that of plateau-specific antibodies (mean ± standard deviation, 23 ± 5 (n = 11) vs. 18 ± 2 (n = 27), P = 0.0106, Mann–Whitney test; Fig. 4b), which may be related to the difference in neutralization breadth and potency between these two groups of antibodies. This finding was not unexpected because the long heavy-chain CDR3, which typically has a major contribution to antigen binding, has been found in several potent and

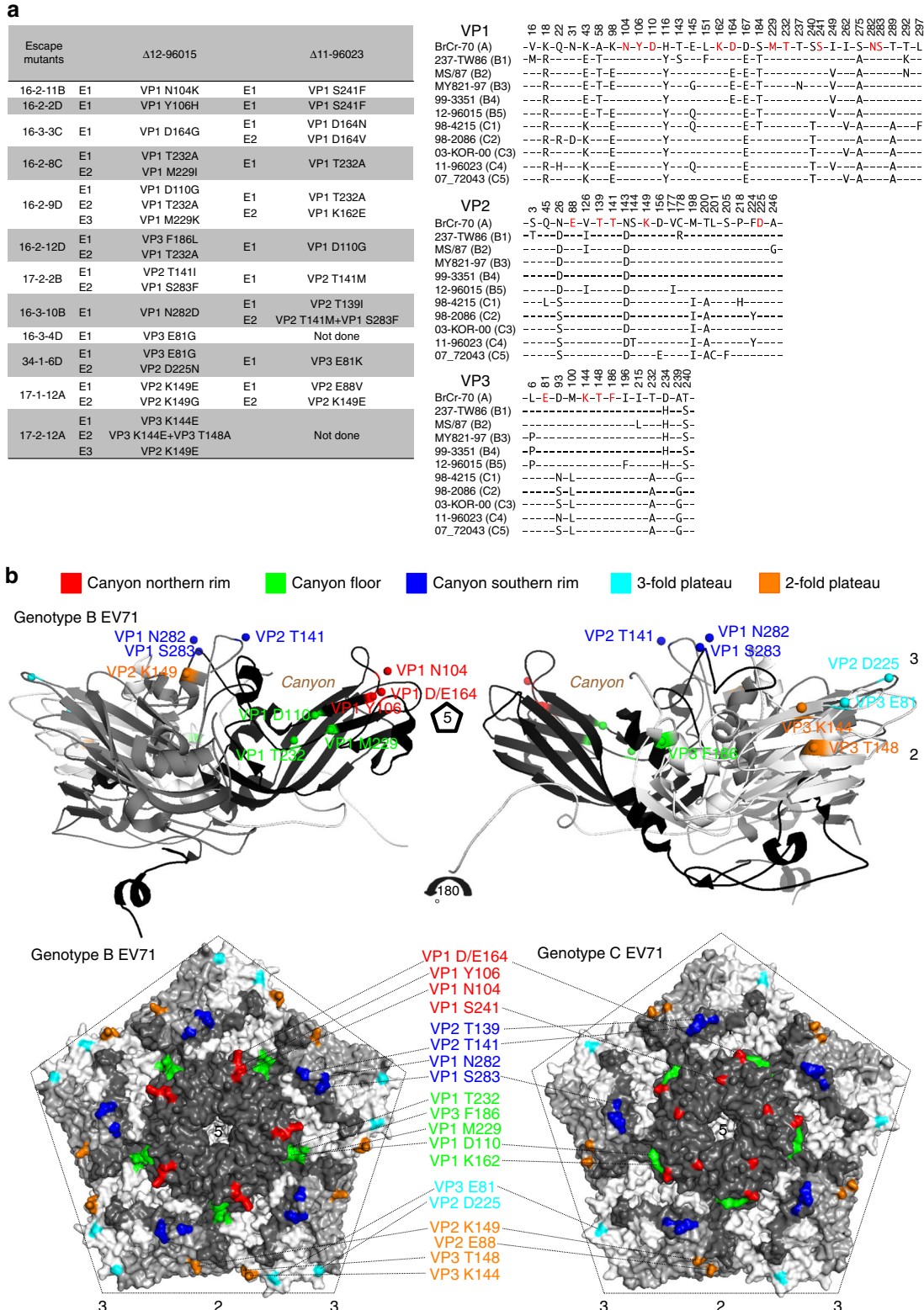

**Fig. 3** Epitope mapping of EV71-neutralizing mAbs. **a** The EV71 12-96015 and 11-96023 escape mutants (*E*) were selected by in vitro propagation with neutralizing mAbs. We identified 25 amino acid substitutions at 19 surface residues of the viral capsid. Antibodies 16-3-4D and 17-2-12A could not neutralize genotype C viruses; thus, we did not perform an in vitro selection of 11-96023 escape mutants for these two mAbs. Residues that are critical for virus neutralization (colored in *red*) are highly conserved among EV71 strains and genotypes. A sequence alignment of the capsid VP1, VP2, and VP3 proteins is shown, and the numbers above the sequence correspond to the amino acid positions in the proteins. **b** Mapping of 19 substitutions selected by representative neutralizing antibodies. The 19 surface residues were mapped to five structural regions of the capsid, designated as the canyon northern rim (*red*), canyon floor (*green*), canyon southern rim (*blue*), 3-fold plateau (*cyan*), and 2-fold plateau (*orange*) epitopes, based on the EV71 structures 3ZFF and 3VBS[24, 25]. The side view of the cartoon diagram of the EV71 icosahedral asymmetric unit and the surface view of the pentamer shown with the 5-fold vertex at the center were created using PyMOL. The capsid VP1 protein is colored in *black*, VP2 colored in *gray*, and VP3 colored in *white*. *5* 5-fold axis, *3* 3-fold axis, *2* 2-fold axis

**Table 3 Cross-reactivity of neutralizing monoclonal antibodies against escape mutants**

| NT against Δ12-96015[a] | VP1 N104K | VP1 Y106H | VP1 D164G | VP1 T232A | VP1 M229I | VP1 D110G | VP1 M229K | VP3 F186L | VP2 T141I | VP1 S283F | VP1 N282D | VP3 E81G | VP2 D225N | VP3 K144E | VP3 K144E + VP3 T148A | VP2 K149E | VP2 K149G |
|---|---|---|---|---|---|---|---|---|---|---|---|---|---|---|---|---|---|
| 16-2-11B | ✖ | - | - | - | - | - | - | - | ↓ | - | - | - | - | - | - | - | - |
| 16-2-2D | - | ✖ | - | - | - | - | - | - | - | - | - | - | - | - | - | - | - |
| 16-3-3C | - | ↓ | ✖ | - | - | - | ✖ | - | - | - | - | - | - | - | - | - | - |
| 16-2-8C | ↓ | - | ↓ | ✖ | ✖ | ✖ | ↓ | - | - | - | - | - | - | - | - | - | - |
| 16-2-9D | ↓ | - | ✖ | ✖ | ✖ | ✖ | ✖ | - | - | - | - | - | - | - | - | - | - |
| 16-2-12D | ✖ | - | ✖ | ✖ | ✖ | ✖ | ✖ | ✖ | - | - | - | - | - | - | - | - | - |
| 17-2-2B | - | - | - | - | - | - | - | - | ✖ | ✖ | - | - | - | - | - | - | - |
| 16-3-10B | - | - | - | - | - | - | - | - | ↓ | - | ✖ | - | - | - | - | - | - |
| 16-3-4D | - | - | - | - | - | - | - | - | - | - | - | ✖ | - | - | - | - | - |
| 34-1-6D | - | - | - | - | - | - | - | - | - | - | - | ✖ | ✖ | - | - | - | - |
| 17-1-12A | - | - | - | - | - | - | - | - | - | - | - | - | - | ↓ | ↓ | ✖ | ✖ |
| 17-2-12A | - | - | - | - | - | - | - | - | - | - | - | - | - | ✖ | ✖ | ✖ | ✖ |

| NT against Δ11-96023[a] | VP1 S241F | VP1 D164N | VP1 D164V | VP1 K162E | VP1 T232A | VP1 D110G | VP2 T141M | VP2 T139I | VP2 T141M + VP1 S283F | VP3 E81K | VP2 E88V | VP2 K149E |
|---|---|---|---|---|---|---|---|---|---|---|---|---|
| 16-2-11B | ✖ | - | - | - | - | - | - | - | - | - | - | - |
| 16-2-2D | ✖ | - | - | - | - | - | - | - | - | - | - | - |
| 16-3-3C | - | ✖ | ✖ | ✖ | - | - | - | - | - | - | - | - |
| 16-2-9D | - | - | - | ✖ | ✖ | ✖ | - | - | - | - | - | - |
| 16-2-8C | - | - | - | ✖ | ✖ | ✖ | - | - | - | - | - | - |
| 16-2-12D | - | - | - | ✖ | ↓ | ✖ | - | - | - | - | - | - |
| 17-2-2B | - | - | - | - | - | - | ✖ | ✖ | ✖ | - | - | - |
| 16-3-10B | - | - | - | - | - | - | - | ✖ | ✖ | - | - | - |
| 34-1-6D | - | - | - | - | - | - | - | - | - | ✖ | - | - |
| 17-1-12A | - | - | - | - | - | - | - | - | - | - | ✖ | ✖ |

NT, neutralization

[a]Neutralizing antibodies are assembled into five subgroups according to the cross-neutralization of EV71 12-96015 and 11-96023 escape mutants: canyon northern rim (16-2-11B, 16-2-2D, 16-3-3C) (red), canyon floor (16-2-8C, 16-2-9D, 16-2-12D) (green), canyon southern rim (17-2-2B, 16-3-10B) (blue), 3-fold plateau (16-3-4D, 34-1-6D) (cyan) and 2-fold plateau (17-1-12A, 17-2-12A) (orange).

✖: Complete loss of neutralization; ↓: More than 8-fold decrease of NT titer; -: maintenance of neutralization

broadly neutralizing human antibodies against human immuno-deficiency virus and influenza virus[37, 38].

Finally, we looked for cross-inhibition of virus binding by the various groups of neutralizing antibodies and compared them to those of the non-neutralizing antibody 17-1-10B (Supplementary Fig. 9). The results showed that while none of neutralizing antibodies were inhibited by the control antibody 17-1-10B, the canyon-specific antibodies cross-inhibited each other in the binding assay. This suggests that, although each of the canyon-specific neutralizing antibodies recognizes a distinct determinant on the northern rim, floor, or southern rim, their footprints may partially overlap. The 3-fold plateau-specific antibodies competed with each other as did the 2-fold plateau-specific antibodies, suggesting that antibodies in the same subgroup share over-lapping footprints. The antibody specific to any of three major epitopes did not compete with the binding of an antibody specific to a different epitope, which indicates that our neutralizing antibodies interact with the canyon, 2-fold, and 3-fold plateau epitopes independently (Supplementary Fig. 9).

**Neutralizing antibody responses in post-infection sera.** Table 1 and Fig. 1 show that the neutralizing antibody responses in donors M and Y were markedly polyclonal, with multiple clones that displayed varying levels of potency and cross-reactivity to EV71 genotypes. The serum screening revealed that day 9 serum from donor M predominantly reacted to canyon epitopes, which suggests that broadly reactive and potent antibody clones emerged and dominated the polyclonal antibody response (Figs. 1 and 5a). In contrast, the serum from donor Y predominantly reacted to the 2-fold plateau epitope, which was compatible with the epitopes recognized by the vast majority of neutralizing clones (Groups 1 and 2 antibodies from donor Y, 86% of neutralizing clones; Tables 1, 3, and Fig. 5a). The EV71-neutralizing clones from donor Z were clustered into one clonal group (Table 1) that targeted the 3-fold plateau epitope, and this focused neutralizing response is also detected at the serological level (Fig. 5a).

A screen of 27 post-infection sera revealed two other pediatric patients, in addition to donors Y and Z, whose sera were predominantly reactive to the 2- or 3- fold plateau epitopes

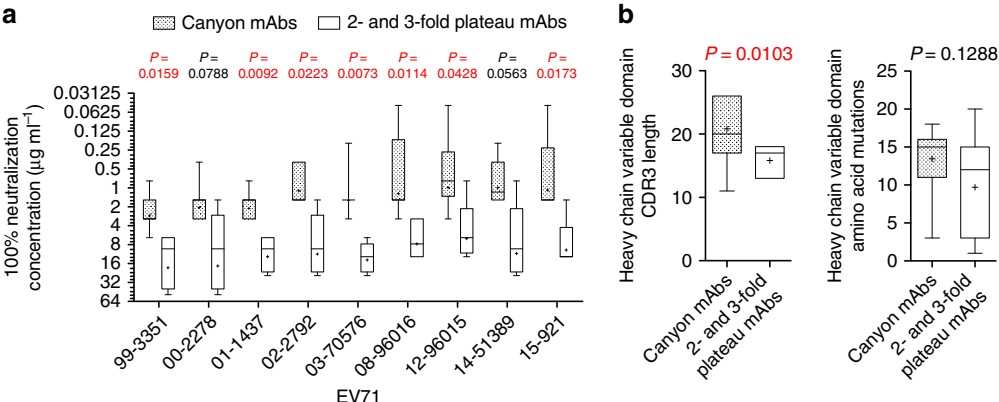

**Fig. 4** Canyon and 2- and 3-fold plateau-specific neutralizing antibodies. **a** Canyon-specific antibodies exhibit more potent neutralizing activities than 2- and 3-fold plateau-specific antibodies. The difference in the neutralization concentrations between two groups (7 or 8 canyon-specific representative mAbs vs. four plateau-specific representative mAbs) was examined by the Mann–Whitney test. The data are shown as *box* and *whisker* plots of the neutralizing activities (µg ml$^{-1}$) of canyon mAbs (*gray dots box*) and plateau mAbs (*empty box*), with the median values shown as *middle bars* and mean values shown as *plus signs*. The neutralizing activity of each representative mAb is defined as the lowest concentration that completely inhibited CPE formation. Each antibody was assayed in triplicate for each virus, and the assay was carried out twice with equivalent results. **b** The average heavy-chain CDR3 length of the canyon-specific antibodies ($n = 11$) is significantly longer than that of the 2- and 3-fold plateau-specific antibodies ($n = 27$) (Mann–Whitney test) (Table 1). The data are shown as *box* and *whisker* plots, with median values shown as *middle bars* and mean values shown as *plus signs*. NT neutralization

(Fig. 5b). The average serum neutralization titer in patients with antibodies focused on these plateau epitopes was significantly lower than the titer of sera that were reactive to both canyon and plateau epitopes (against 12-96015, $P = 0.0200$; against 11-96023, $P = 0.0188$, Mann–Whitney test) or predominantly reactive to canyon epitopes (against 12-96015, $P = 0.0334$; against 11-96023, $P = 0.0365$, Mann–Whitney test; Fig. 5b). These results supported the presence of epitope-associated neutralization in the polyclonal antibodies induced by natural EV71 infection in children.

Of the 27 post-infection sera, 18 (67%) showed a $\geq$ 8-fold reduction in neutralization titers against at least one mutated neutralization epitope, and 13 (48%) were sensitive to two or more mutated epitopes. We noted that the highest proportion of tested sera (13 out of 27, 48%) were sensitive to canyon northern rim mutations, and the lowest proportion (4 out of 27, 15%) were sensitive to canyon floor mutations (Fig. 5c), which implies that the epitope at the bottom of canyon may be relatively less immunogenic for eliciting a neutralizing antibody response in naturally infected children.

**Pre- and post-attachment neutralization by mAbs.** To identify the stage of infection at which the antibodies neutralize EV71, we performed pre- and post-attachment neutralization assays. The results showed that pre-incubation with neutralizing antibodies prevented subsequent viral infectivity of RD cells in a dose-dependent manner (Fig. 6). The canyon rim-specific antibodies 17-2-2B and 16-3-10B had the most potent inhibitory activities at the pre-attachment step and the 2-fold plateau-specific antibody 17-2-12A had the poorest inhibitory activity.

We also tested whether the antibody neutralized infection at the post-attachment step. The virus was allowed to bind RD cells at 4 °C for 1 h. Then, virus-attached cells were treated with serially diluted mAbs. We found that only canyon rim-specific antibodies showed fairly good inhibitory activity at the post-attachment step (Fig. 6). Neutralization at this stage required 10- to 100-fold higher concentrations of the canyon floor, 2-, and 3-fold plateau-specific antibodies than the EC$_{50}$ at the pre-attachment stage.

## Discussion

Our in-depth analysis of the EV71 plasmablast-derived antibody repertoires from three infected children demonstrated

epitope-associated neutralization potency and breadth at the serological and clonal levels. Moreover, a polyclonal antibody response focused on the margin of the EV71 capsid pentamers was detected in donors Y and Z, and this response was associated with low neutralization titers. In contrast, the strong, cross-neutralizing antibody response detected in donor M was composed of up to eight variable domain-related groups of neutralizing clones, the majority of which (7 out of 8 groups) recognized overlapping regions on the capsid canyon and displayed potent activities. The mechanism by which the specificity of the neutralizing antibody repertoire becomes focused was unclear; however, there are several possibilities, including (i) clonal selection of B cells from a pre-existing immune repertoire shaped by past exposure to related virus[39–41], (ii) expansion of B cells that influence antigen processing and presentation to T cells in a reciprocal interaction[42, 43], (iii) selection of B cells with high affinity to intrinsically attractive epitopes[44, 45], or (iv) a skewed response during the first exposure to a new virus[46]. These mechanisms are not mutually exclusive and may act together.

The influence of physiological age-related variation on the clonality and specificity of the EV71-specific antibody response should be trivial in our donors, because older children, unlike new-borns and infants, have developed an efficient adaptive immune system for responding to T cell-dependent antigens[13, 47–49]. Our clonal analysis of EV71-neutralizing antibodies revealed eight variable domain-related groups of canyon-specific antibodies and four groups of plateau-specific antibodies; this evidence of polyclonality points to the abundance of the repertoire available for recognizing functional epitopes on the EV71 capsid in older children. Further evidence for immune maturity came from the robust virus-specific IgG plasmablast responses and extensive somatic mutations in the variable domains, suggesting rapid recall of EV71-specific B cell responses in these donors.

Our data suggest that the highly focused antibody responses to plateau epitopes in genotype B5-infected children result in greatly reduced neutralization titers against genotype C4 EV71. This genotype of EV71 has re-emerged, replacing the previously predominant genotype B5, and it has been frequently isolated in the latest outbreak in Taiwan in 2016[6, 7, 15]. This implies that these children with plateau epitope-focused antibody repertoires may

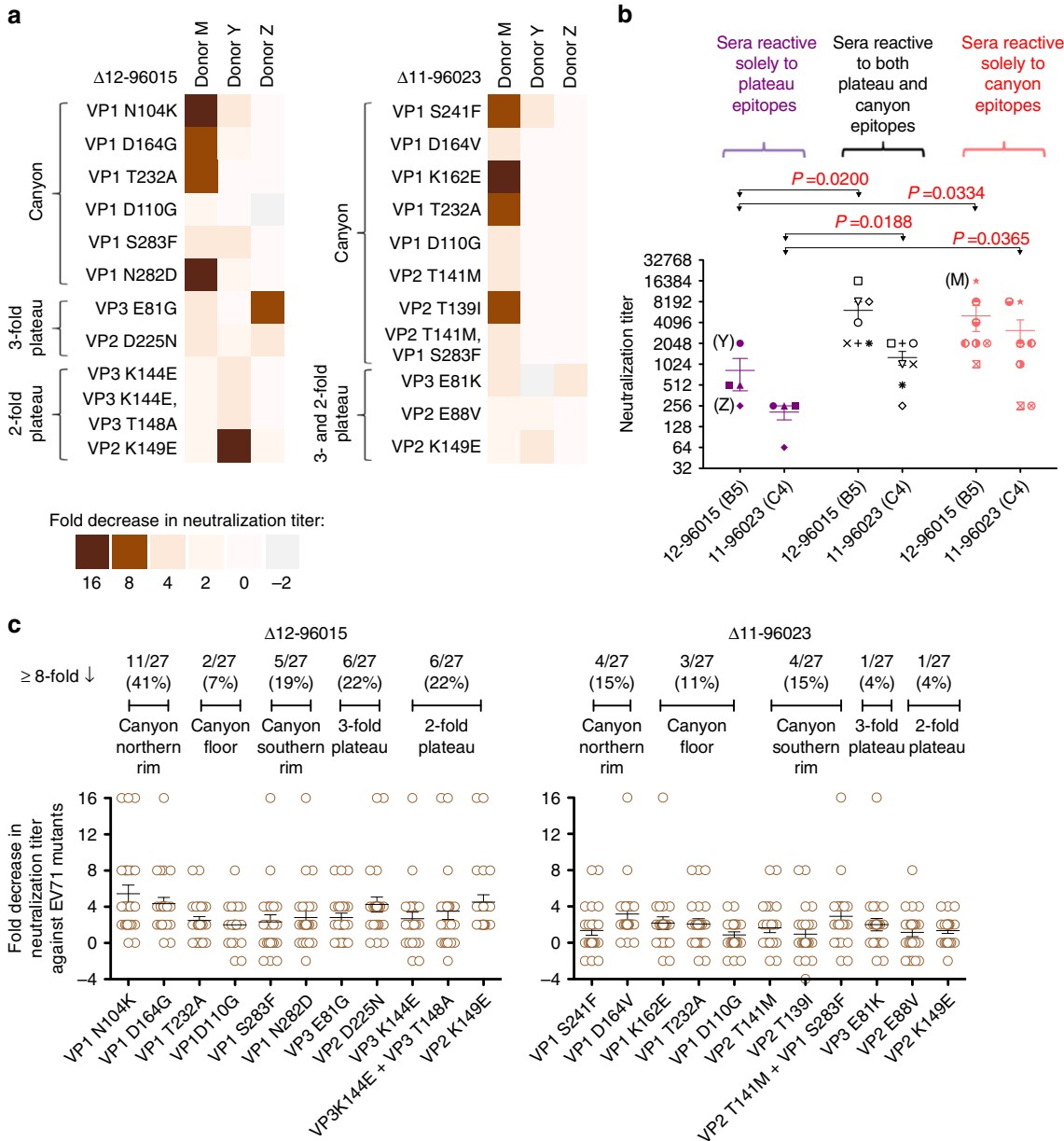

**Fig. 5** Serological responses to natural EV71 infection in children. **a** Post-infection sera from three mAb donors (day 9 serum from donor M and day 11 sera from donors Y and Z) were tested for neutralization of wild-type and escape mutants of EV71. **b** Post-infection sera from pediatric patients with laboratory-confirmed EV71 infection were tested against the wild-type virus and escape mutants. The figure shows a comparison of the neutralization titers against EV71 12-96015 and 11-96023 between two sera groups (Mann–Whitney test). The symbols in the figure represent one particular subject (i.e., *solid circles* are donor Y and *solid diamonds* are donor Z). **c** Of the 27 post-infection serum samples, 16 (59%) and 7 (26%) showed a ≥ 8-fold reduction in neutralization titer against at least one 12-96015 and 11-96023 escape mutant, respectively. The percentages of samples that were reactive to each epitope are shown. *Black lines* represent the mean ± standard error of the mean. All samples were tested in duplicate from two independent experiments

have an increased risk of EV71 infection, to which they lack effective antibody immunity, and they may be more likely to develop severe illness and continue to shed viruses so that these viruses can spread to other more susceptible hosts, such as infants.

Two human serum studies have characterized the EV71-specific antibody response using a multi-genotype EV71 panel and mutants with introduced substitutions. The authors suggested that several capsid residues which vary among different genotypes might serve as antigenic determinants; however, their results showed that none of these substitutions individually affected the activity of polyclonal antibodies[16, 36]. The majority of the canyon- and plateau-specific antibodies that we detected

recognize capsid residues that are conserved across virus genotypes. However, we found that specificity to the plateau region was associated with limited neutralization breadth and significantly lower potency. We also showed that the EV71 capsid-specific neutralizing antibodies in a subset of children predominantly reacted to plateau epitopes to such an extent that they had a significantly reduced neutralization titer (Figs. 1 and 5). These results suggest that a detailed clonal analysis of the virus-specific plasmablasts induced by recent exposure could be a valuable supplement to traditional serological approaches for dissecting the breadth and specificity of the neutralizing antibody repertoire. Nevertheless, we have to mention that some children with low neutralization titers may not always have antibodies

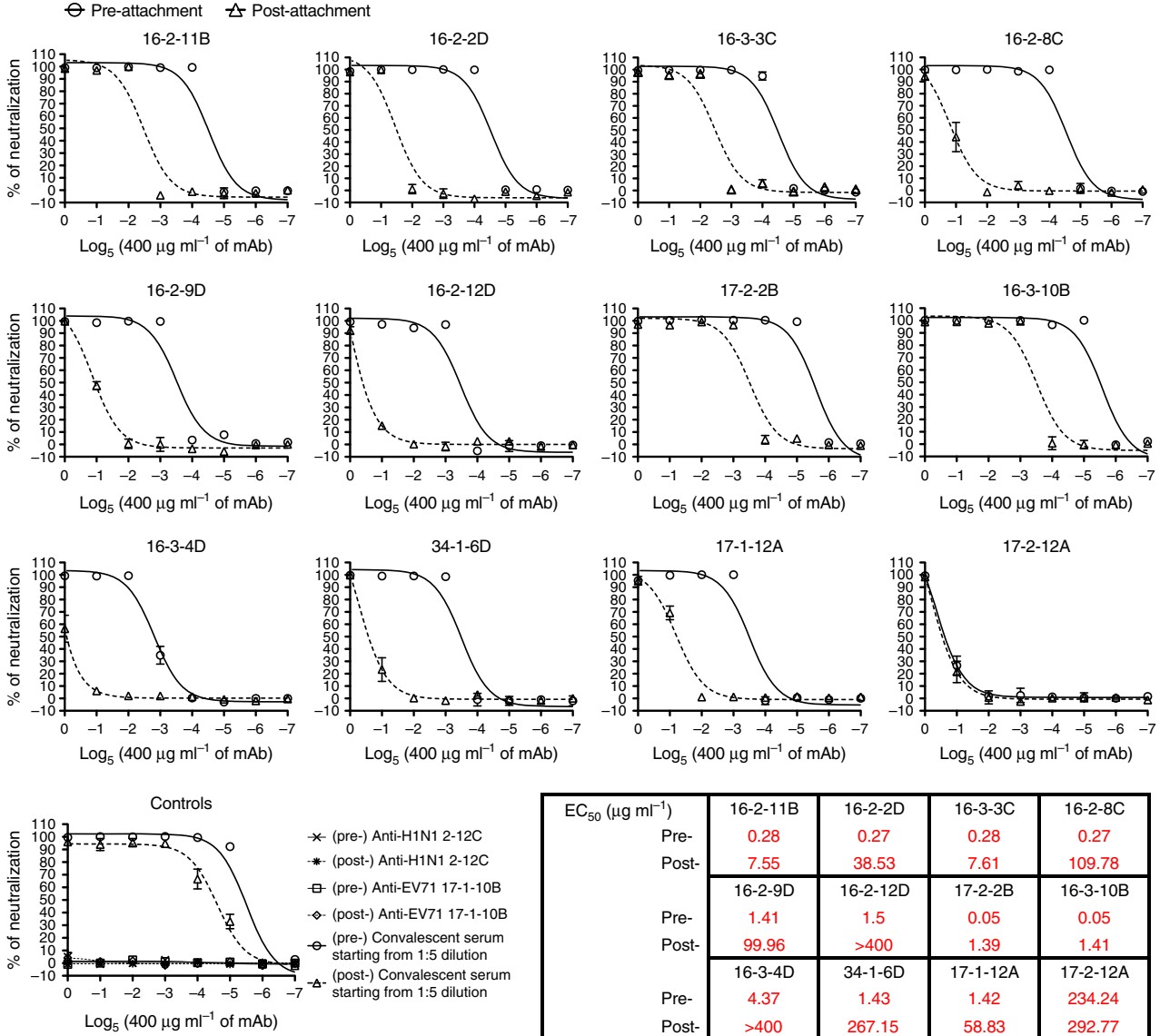

**Fig. 6** Inhibition of EV71 infectivity at the pre- and post-attachment stages by neutralizing antibodies. At the end of the incubation period, the cells were examined for the cytopathic effect. In addition, cell viability was determined by the MTT (3-(4,5-dimethylthiazol-2-yl)-2,5-diphenyl tetrazolium bromide) assay, and the percentage of neutralization for each treatment was calculated. The data are the mean ± standard error of the mean from two independent experiments. $EC_{50}$ 50% effective concentration

focused on the plateau epitopes identified in the study (Fig. 5b), as the EV71 capsid epitopes mapped by our neutralizing mAbs may not be relevant in 9 out of 27 (33%) of the post-infection serum samples screened in our study, and the neutralization epitopes targeted in these sera are still unclear.

Our study has several limitations. First, the antigen-specific plasmablasts that we isolated to make the mAbs were elicited during acute infection with a particular strain of EV71; therefore, the antibody repertoire may be enriched for this recent antigen. Second, IgM and IgA plasmablasts were not used to generate mAbs in the study. Upon acute EV71 infection, virus-specific IgG plasmablast responses are predominantly detected in children >3 years old; however, the IgG responses are sometimes accompanied by nearly equivalent IgM plasmablast responses[15]. Thus, the neutralization breadth and potency of the EV71-specific IgM and IgA antibody clones in our donors are unknown. Third, plasmablast-derived antibody clones may not represent the complete repertoire of B cells specific for EV71. Previously, we showed that the virus-specific plasmablast response in peripheral blood peaks 4–7 days after illness onset, and there is a correlation between cell frequency and serum titer[15]. Therefore, in this study, day 7 plasmablasts were utilized to construct the EV71-specific monoclonal antibody repertoire. Other approaches, i.e. construction of phage display libraries and immortalized memory B cells from convalescent donors, have been used to isolate the antigen-specific B cells that were generated several years earlier and maintained in the memory cell pool[50]. However, the importance of memory B cells in protection from re-infection and the contribution of memory B cells to serological antibody levels are controversial[51–53]. Phage display of random heavy and light-chain gene pairs may generate a more diverse pool of antibodies, but this library of antibody fragments may not faithfully represent the physiological immunoglobulin gene pairs[50]. Recently, high-throughput sequencing of the gene rearrangements encoding B cell immunoglobulins has been showed to be another useful tool for identifying the clonal expansion signatures that are correlated with the magnitude and specificity of the serological response to pathogens and vaccination[54].

Novel neutralization epitopes were identified in the canyon region of the capsid. The canyon northern rim epitope is located at the border of the 5-fold mesa, as shown in Supplementary Fig. 10a, and it overlaps or is in close proximity to the binding site for heparan sulfate and P-selectin glycoprotein ligand-1, two proposed attachment receptors on the surface of eukaryotic cells[55, 56]. The canyon northern rim epitope is also adjacent to VP1 residue 172, one of the binding sites for human scavenger receptor class B member 2 (hSCARB2) (Supplementary Fig. 10a)[57]. The cellular receptor hSCARB2, which is expressed on a wide range of cell types, including human neurons and RD cells, facilitates EV71 infection by contributing to viral attachment, entry, and subsequent uncoating[58]. The EV71-neutralizing antibodies probably inhibit virus attachment and interfere with virus-hSCARB2 complex formation.

The mouse monoclonal antibody 22A12 recognizes VP1 residues on the canyon southern rim (Supplementary Fig. 10b)[28, 59], but poorly neutralizes the virus inoculum because it can be sequestered by empty capsids. We did not observe a similar situation with our neutralizing antibodies that recognize distinct sites on the canyon southern rim. The canyon southern rim epitope overlaps the hSCARB2 binding sites (Supplementary Fig. 10b)[60], and it has been shown that antibody interference with virus-hSCARB2 interactions can lead to neutralization[27].

Canyon floor-specific antibodies are rather unusual in picornavirus-specific antibody responses. On the basis of the crystal structure and epitope studies of human rhinovirus 14 in the 1980s, a hypothesis was proposed that the capsid canyon, a deep surface depression surrounding each 5-fold mesa, is inaccessible to large antibodies because this canyon region is an invaginated structure[61]. However, the capsid canyon of EV71 is shallower than that of other enteroviruses, which may allow antibodies to reach the deepest part of the structure[24]. VP1 residues 229 and 232 of the canyon floor epitope are involved in the proposed 'adaptor-sensor region', which undergoes drastic conformational movement following virus attachment to receptor, and such movement is supposed to be a prelude to the expulsion of pocket factor, virion expansion, and uncoating[24]. The binding of a canyon floor-specific antibody may prevent this conformational change thus stabilize the capsid or trigger premature uncoating[25]. Further structural analysis of antibody-EV71 complexes will help to elucidate the interaction between antibody and this adaptor-sensor region.

In contrast to canyon epitopes, 3- and 2-fold plateau epitopes are not within or adjacent to known receptor binding sites, although antibodies targeting these epitopes neutralize viral infectivity prior to virus attachment. It is possible that these epitopes are in the neighborhood of as-yet unidentified receptor-binding sites;[62] however, this neutralization may involve an alternative mechanism[63, 64].

The plateau-specific antibodies bind to both genotypes B and C, but most (3 out of 4 clonally related groups) of these antibodies failed to neutralize or only weakly neutralized genotype C viruses. Although the sequences of the plateau epitopes are conserved among EV71 strains, different side chain orientations and electrostatic potentials have been observed at key residues (Supplementary Fig. 11) in previously published capsid structures[24, 25]. Conformational differences in the key residues of the conserved epitope have been reported to affect the interaction between neutralizing antibody and virus[38]. For example, the influenza HA stem antibodies CR6261 and F10 neutralize group 1 influenza viruses, but fail to neutralize group 2 viruses. This lack of neutralization is thought to be due to the unfavorable orientation of a highly conserved tryptophan within the stem epitope (HA2 residue 21) of group 2 viruses, which causes steric hindrance with antibody binding[38]. The three-dimensional structure would be useful for examining the role of EV71 plateau epitope conformational variability in the resistance to broad neutralization.

There is no specific therapy to prevent or arrest the progression from mild hand-foot-and-mouth disease to severe neurological disease, although anecdotal experience has suggested the effectiveness of early intravenous immunoglobulin administration[1]. Clinical experience with passive therapy in immunocompromised patients supports the importance of neutralizing antibodies in the treatment of severe enteroviral infections[65]. Here, we presented a panel of broadly neutralizing EV71-specific mAbs, the majority of which display potent activity against multi-genotype EV71, including recently circulating strains. Potent antibodies can neutralize viruses at concentrations <100 ng ml$^{-1}$ and can inhibit infectivity prior to viral attachment at <50 ng ml$^{-1}$. This activity is comparable to that previously reported for human mAbs against influenza, respiratory syncytial, and dengue viruses[19, 33, 38–40, 66]. Our best-in-class human mAbs were 10- to 100-fold more potent than the previously reported murine mAbs[27–31] and were comparable to two murine mAbs that recognize the 5-fold vertex and the epitope crossing the interface of capsid protomers[32, 62]. These findings warrant further investigation into the prophylactic and therapeutic efficacies of neutralizing antibodies in small and large animal models and the potential clinical use of such antibodies in the near future.

## Methods

**Ethical approval**. The study protocol and informed consent were approved by the ethics committee at the Chang Gung Medical Foundation Institutional Review Board. Each subject provided written informed consent. The study, and all associated methods, were carried out in accordance with the approved protocol, the principles outlined in the Declaration of Helsinki, and Good Clinical Practice guidelines.

**Production of human mAbs**. Peripheral blood mononuclear cells (PBMCs) were obtained from three pediatric patients in Taiwan with laboratory-confirmed genotype B5 EV71 infection. The PBMCs ($1 \times 10^6$) were stained with a mix of fluorescent-labeled antibodies to cellular surface markers, including PB anti-CD3 (5 µg ml$^{-1}$, clone UCHT1, BD), FITC anti-CD19 (1:10 dilution in a 100 µl experimental sample, clone HIB19, BD), PE-Cy7 anti-CD27 (1:20 dilution in a 100 µl experimental sample, clone M-T271, BD), APC-H7 anti-CD20 (5 µg ml$^{-1}$, clone L27, BD), and PE-Cy5 anti-CD38 (1:10 dilution in a 100 µl experimental sample, clone HIT2, BD). Plasmablast subsets were identified by flow cytometry, and cognate heavy- and light-chain variable domain genes were isolated from the sorted single plasmablasts (Supplementary Methods)[15, 19]. The genes were cloned and expressed in 293T cells (ATCC, CRL-3216) in serum-free transfection medium. Monoclonal antibody-containing culture supernatant was collected after 5 days. The antibody yield was determined by IgG-ELISA. Serial dilutions of a human IgG of known concentration (Sigma) were prepared to generate a standard curve. The standard curve was used to calculate the concentration of antibody in supernatants based on their OD value. Supernatants containing full-length IgG1 human mAb were further analyzed to determine antigen specificity and neutralization activity.

A panel of neutralizing and non-neutralizing human mAbs were further expanded and purified, and the purity was assessed by SDS-PAGE. The concentration of the purified IgG1 mAbs was determined by measuring the absorption at 280 nm (A$_{280}$).

**Viruses**. Several EV71 clinical strains isolated in 1998–2016, including 98-2086, 98-4215, 99-1691, 99-3351, 00-2278, 01-1437, 02-2792, 03-70576, 04-72232, 05-1956, 07-72043, 08-96016, 10-96018, 11-96023, 12-96015, 14-51389, 15-921, 16-50444, and 16-50555, were used for experiments. Other enteroviruses, including 12-50891 (coxsackievirus A2), 14-2060 (coxsackievirus A16), 15-50909 (coxsackievirus A16), and 14-2795 (enterovirus D68), were used as controls. All viruses were obtained from the Clinical Virology Laboratory of Chang Gung Memorial Hospital (Contract Laboratory of the Taiwan Centers for Disease Control), plaque purified, and amplified by RD cells (ATCC, CCL-136) cultured in virus growth medium (DMEM/2% FBS/penicillin and streptomycin). Infected cells with a complete cytopathic effect (CPE) and culture supernatants were collected, freeze-thawed three times, and centrifuged. Virus-containing supernatants were collected, and the viral titer was determined by a TCID$_{50}$ assay with RD cells using the Reed-Muench method. EV71 viral RNA was extracted, and the P1 region was amplified with a set of primers (Supplementary Table 3). The P1 region, encoding the proteins VP1–4, was sequenced, and the genotype of the EV71 strains was analyzed based on the VP1 sequence.

Purified 12-96015 and 11-96023 EV71 were prepared according to a previously reported method[67], except that RD cells instead of HeLa cells were used to produce large amounts of virus-containing supernatant. Once a complete CPE was observed, the cells and supernatants were collected, freeze-thawed three times, and centrifuged. We concentrated the virus-containing supernatants by polyethylene glycol precipitation and high-speed centrifugation. Then, the pellet was resuspended and incubated with DNase. The reaction was stopped by adding EDTA, and the supernatant was clarified by centrifugation. The virus was pelleted through a sucrose cushion and resuspended. After centrifugation, virus-containing supernatants were layered on a 10–35% potassium tartrate gradient for final purification by ultracentrifugation. Both the upper and lower bands of virus were separately collected, and then the potassium tartrate was removed by ultrafiltration. Viral density was measured by a BCA protein assay.

**Flow cytometry-based binding assay**. To prepare EV71-infected RD cells, a confluent monolayer of RD cells were incubated with the optimized infectious dose of virus 1 day before the experiment. The next day, the cells were collected, washed, and resuspended. Fixed and permeabilized cells were blocked with saponin-3% BSA. Cells were probed with primary antibodies (mAb-containing cell culture supernatant with saponin, purified mAbs, or serum prepared in BD Perm/Wash™ buffer; mAb, 5 µg ml⁻¹; serum, 1:125 dilution) and mouse IgG mAbs to EV71 3C (1 µg ml⁻¹, GeneTex). Bound primary antibodies were detected with fluorescence-conjugated anti-IgG secondary antibodies (Goat anti-human IgG, 2.5 µg ml⁻¹; Goat anti-mouse IgG, 1.5 µg ml⁻¹) (Thermo Fisher Scientific) in BD Perm/Wash™ buffer. Cells were analyzed with a BD FACSCanto™ II flow cytometer. Results were derived from an analysis of 10,000 gated events of EV71-infected (3C-positive) cells and are shown as the percentage of EV71-infected cells that bound human anti-EV71 antibodies. Mock-infected RD cells were used as an antigen control, and non-transfected cell culture supernatant, an anti-influenza human monoclonal antibody (2–12C, prepared in house)[19], and an anti-EV71 VP2 mouse monoclonal antibody MAB979 (EMD Millipore) were used as antibody controls for each experiment.

**ELISA**. The purified virus preparations were absorbed to the wells of a microtiter plate. Nonspecific binding was blocked with 3% BSA in PBS. The mAb-containing cell culture supernatant (5 µg ml⁻¹), purified antibody (5 µg ml⁻¹), or serum (1:125 dilution) was applied, and bound virus-specific antibodies were detected with an HRP-conjugated anti-human IgG secondary antibody (0.5 µg ml⁻¹, Rockland).

**Immunoprecipitation and western blotting**. For immunoprecipitation, Protein G Dynabeads® (Thermo Fisher Scientific) were prepared according to the manufacturer's protocol and incubated with the mAb preparation (5 µg ml⁻¹). The bead-antibody complex was incubated with pre-cleared EV71-containing supernatants. The eluate was resolved by SDS-PAGE and analyzed by silver staining (Thermo Fisher Scientific) and western blotting (Supplementary Methods).

To examine the EV71-specific mAb-binding sites by western blotting, EV71-containing supernatants from RD cells were processed under heated/reducing and unheated/non-reducing conditions, separated by SDS-PAGE, and transferred to a nitrocellulose membrane. After blocking, the membrane was probed with the mAb preparation (5 µg ml⁻¹), and an HRP-conjugated anti-human IgG secondary antibody (1.25 µg ml⁻¹, Dako) was used to detect bound antibodies.

**Neutralization assay**. The mAb-containing supernatants, purified mAbs, and sera were tested for neutralizing activity against plaque-purified EV71, EV71 escape mutants, or EV71 collected from transfection with infectious clones. Neutralizing activity was evaluated by a cell-based neutralization assay[13, 15]. Sera were pre-treated at 56 °C for 30 min. Then, an aliquot (50 µL) of each antibody preparation was mixed with an equal volume of 100 TCID₅₀ virus per well in a 96-well plate and incubated at 37 °C for 2 h. Then, 100 µL of an $8 \times 10^4$ RD cell suspension was added to each well and incubated at 37 °C for 5 days. For each experiment, the cell control, positive serum/antibody control, and virus back-titration were set up. All samples were assayed in triplicate. At the end of the incubation period, cells were fixed with 5% glutaraldehyde and stained with 0.1% crystal violet. When a CPE was observed at 1 TCID₅₀ in the control wells of the virus back-titration, the neutralization titer was determined as the reciprocal of the sample dilutions that completely prevented the cytopathic effect in all three triplicate wells.

**Generation of escape mutants**. Wild-type 12-96015 (GenBank accession number KX267854) and 11-96023 (GenBank accession number KX267855) EV71 were incubated with excess mAb for 1 h and then inoculated into an RD cell monolayer (Supplementary Methods). The inoculated cells were incubated at 37 °C for 4 days. The CPE was normally observed in the first or second cycle of re-infection, and hence mAb-resistant mutants developed. The plaque-purified viruses were confirmed as mAb-resistant mutants by a lack of mAb binding and neutralization activity. The P1 region of the escape mutants was sequenced (Supplementary Table 3), and the nucleotide and amino acid sequences were compared to those of the parent virus.

**Site-directed mutagenesis**. The sequence of an infectious cDNA clone of genotype B5 EV71 strain N1745-TW08 (GenBank accession number KT354870.1)[36] was altered by site-directed mutagenesis (Supplementary Methods). Primers were designed and nucleotide substitutions were generated on plasmids using the QuikChange Site-directed Mutagenesis Kit (Agilent). Linearized viral DNA was transcribed in vitro, and the obtained RNA was transfected into RD cells with Lipofectamine 2000 (Thermo Fisher Scientific). The introduction of a single amino acid substitution was confirmed by sequencing.

**Pre- and post-attachment neutralization assay**. For the pre-attachment neutralization assay, serially diluted antibodies were incubated with 100 TCID₅₀ of virus at 37 °C for 1 h. Then, the pre-chilled antibody-virus mixture was added to a confluent monolayer of RD cells plated the day before and incubated at 4 °C for 1 h. After washing with cold virus dilution medium to remove unbound virus, the cells were incubated at 37 °C for 5 days. For the post-attachment neutralization assay, 100 TCID₅₀ of pre-chilled virus was incubated with a confluent monolayer of RD cells plated the day before at 4 °C for 1 h. After washing with cold virus dilution medium to remove unbound virus, the cells were immediately incubated with serially diluted antibodies at 37 °C for 1 h. After washing, cells were incubated at 37 °C for 5 days. At the end of the incubation period, the viability of cells was determined by the MTT [3-(4,5-dimethylthiazol-2-yl)-2,5-diphenyl tetrazolium bromide] assay (Supplementary Methods).

**Statistics**. The TCID₅₀ for each virus in the RD cell line was calculated by the Reed-Muench method using SPSS. The EC₅₀ for each neutralizing antibody in the pre- and post-attachment assays was calculated by nonlinear regression analysis using GraphPad Prism. The differences in the number of somatic mutations in the variable domains, CDR3 lengths, and neutralization concentrations between two groups were examined by the Mann-Whitney test using GraphPad Prism. A $p$ value less than 0.05 was considered statistically significant. Graphs were generated with Microsoft Excel and GraphPad Prism.

**Data availability**. The sequences of all the representative antibodies have been deposited in GenBank (Accession Numbers KY354551–KY354578). The data that support the findings of this study are available in the article and its Supplementary Information files, or from the corresponding author upon reasonable request.

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

## Acknowledgements

We acknowledge the BD FACSAria™ cell sorter service provided by the Core Instrument Center of Chang Gung University. This work was supported by Chang Gung Medical Research Program (CMRPG3E1471-3) and Ministry of Science and Technology of Taiwan (MOST 103-2314-B-182A-116, MOST 105-2314-B-182A-142, and MOST 105-2321-B-182A-004-MY3).

## Author contributions

K.-Y.A.H.: Conceived and designed the study. K.-Y.A.H., M.-F.C., Y.-C.H., S.-R.S., C.-H.C., J.-J.L., J.-R.W., and K.-C.T.: Carried out the experiments. K.-Y.A.H., M.-F.C., Y.-C.H.,

S.-R.S., C.-H.C., J.-J.L., J.-R.W., K.-C.T., and T.-Y.L.: Performed the data analysis and figure/table preparation. K.-Y.A.H., M.-F.C., S.-R.S., and T.-Y.L.: Wrote the manuscript.

## Additional information

**Competing interests:** All authors declare no competing financial interests.

