## [Peer Review File · Nature Communications]

Reviewers' Comments:

Reviewer #1:

Remarks to the Author:

This paper represents a great deal of work on the part of the authors, who have generated monoclonal antibodies from plasmablasts of 3 children infected with enterovirus 71, tested the neutralizing capacity of the antibodies, and selected viral escape mutants to map the epitopes recognized by a representative subset. The results indicate that effective neutralizing antibodies recognize epitopes associated with the rims or the base of the viral canyon, and are effective against viruses of diverse genotypes; in contrast, antibodies that bind plateaus near the 2-fold and 3-fold axes neutralize weakly. The authors further test sera from a number of other infected children against the panel escape mutants, and are able to infer that some children have responses focused more on the canyon region or on the plateaus; they conclude that those whose responses are focused on the plateaus have weaker and less broadly protective neutralizing responses.

I think that, with few exceptions, the experiments are well-designed and performed, and I think the paper provides interesting new information about the human neutralizing response to EV71 infection. However, I think the paper needs extensive revision for clarity, and I am not convinced that there is a subset of children with weak neutralizing responses focused on the plateaus. I also find that some of the authors' statements about the importance of the work, and its relevance to vaccine design, etc. are unconvincing and need better justification.

Substantive issues.

1. I don't think that it is appropriate to conclude from the data presented here that, in general, children with lower neutralizing titers have antibody responses focused on the plateaus, or that children with responses focused on the plateaus have a lower neutralizing titer, as is stated in the discussion (Lines 351-353) and in the abstract. As far as I can see, the data show that in a single child with low titer, the response was focused on plateau epitopes as determined from the repertoire of monoclonal antibodies. In a group of children whose polyclonal responses were focused on the plateau epitopes, there was less activity against a heterologous virus, but the neutralizing titers against B5 were the same in both groups of children (as shown in Figure 5B). And speaking of 5B, what are the "controls" shown in the third column?

2. Figure 2B. The effective dilutions of monoclonal antibody supernatants can't be used to compare antibodies unless we know that the antibody concentrations are the same. Reading the methods describing the binding curves I'm not sure what was measured here. Counting the number of infected cells has nothing to do with the binding avidity of the antibody. I think there was a known number of infected cells (determined how?) and this was counted as 100%. So the percentage is the number of infected cells detected (as a percentage of the total infected cells) rather than the number of infected cells (as a percentage of the total cells).

I don't understand the statement (lines 166-168) discussing 2B. To me, "while" means "at the same time." And I see no reason why the number of infected cells detected with a particular antibody should relate to its neutralizing efficacy.

3. Donor Z had a low neutralizing titer, and monoclonals generated from his cells had weak activity against the genotype B5 virus used for mapping epitopes. As far as I know, the various genogroups of EV71 are determined on the basis of sequence rather than on serological properties. Do we know that all B5 viruses are serologically the same? Is there any possibility that this child was infected with a B5 variant that was serologically distinct from the virus isolate tested in the lab? Was the child's virus sequenced in its entirety? Maybe he had a terrific response against the virus with which he was really infected. Can we exclude the possibility that it was a recombinant in which the EV71 neutralizing epitopes were replaced by sequences from another virus. This may seem far-fetched, but the authors are placing a lot of weight on the results from

this single child.

4. Do we know that plasmablasts isolated at day 9 (or serum taken at day 11) reflect the breadth of the antibody response that will ultimately determine protection from reinfection? Does the response continue to mature after that, with selection for higher-affinity clones?

5. I don't think the experiments with antibodies added before and after attachment are easy to interpret (and I'm not sure what the authors really make of them). All the antibodies inhibit in both situations, and virtually all of them show reduced activity after virus has already bound. The weaker antibodies do worse in both cases. I don't think that these experiments really can be used to support the idea that some antibodies directly block interaction with receptors and others do not. Antibodies could cause virus clumping, they could stabilize viruses against uncoating, they could trigger premature uncoating. More evidence would be needed to distinguish among the many possibilities

6. In one place it's said that the 2 most potent antibodies were effective at 24 and 98 ng/ml, but something slightly different is shown in 2C. If the specific numbers aren't important, why be so specific?

7. In the mapping of the epitopes, you must have sequenced the genomes of escape mutants, but sequencing is mentioned nowhere.

8. Paragraph beginning 203 needs more information and rewording to be meaningful.

9. In Figure 3A, are the different residues involved in escape from each particular antibody (for example VP1 M229 and VP2 T232, or VP3 E81 and VP2 D225) located close to each other on the capsid surface?

Issues of presentation:

The paper needs careful review by a professional editor to correct numerous small errors in grammar and diction— especially the use or omission of articles and the choice of plural vs. singular. The tenses should be brought into accord with the general convention for scientific papers, in which the present tense is used to describe generally accepted facts, published results, and major conclusions, whereas the past tense is used to describe the results of specific experiments described in the results section. In many cases these glitches are just an annoyance to the reader, but in some cases they are truly confusing. I list some of these below, but there are many more.

In a number of places the text contained in the figures is very small and difficult to read. For example, the table in 2C could be split off as a table and made larger, the sequences in 3A and the labeling of residues in 3B are virtually illegible. In Fig 5A it would be helpful to indicate the location of specific residues, as they are marked or colored elsewhere; in the key to the figure "fold decrease..." is tiny, and the dark brown color hides the numbers (?8 and 16?).

Examples of presentation problems:

The title is uninformative. "Phenomena" makes it sound like the paper is about random observations rather than a scientific inquiry.

Abstract: I don't think there is such a thing as an "endemic." What is a "multi-genotype" virus? It's not obvious to what implications the work has for evaluation of vaccines or antibody therapeutics. Vaccines will be tested empirically for generation of neutralizing responses and protective efficacy.

Introduction: Line 44, outbreaks "are" associated with. Line 58, "alternative" to what? Line 68 "mouse" or "murine." Line 92 "unravel the antigenic landscape" Huh?

Results: Line 102 referred to AS donors... Average age is silly, given that there are only 3 subjects. Line 109. Paired sera.

Line 114. Each donor developed "distinct" antibody response. In what way?

Line 117. small-titer increase? ?small increase in titer?

Who are donors B and C? And why would patients exposed to one virus mount a response to another virus. Rephrase— ? the antibody produced in response to infection neutralized B5 but failed to neutralize...

Line 134. We first observed an efficient development of neutralizing clones in 3 donors.. I don't understand the use of "first" and I don't understand the repeated use mention 3 donors. Were there other donors in whom you didn't observe this? Do you mean 'all 3 donors?'

Line 149, 152. repertoires WERE;

Lines 159 and 160. I have no idea what this means. Please explain

Line 163. map the relevant epitopes.

Line 174. broadly reactive

Line 181. "Confirmed"? Is this the confirmation of someone else's result? Of a result you discussed earlier in the paper?

Discussion:

Line 308. "responsible." This is a supposition. What you know is they were associated.

Line 311: 'focused on the overlapping regions on the capsid canyon with potent activities.' Makes no sense as written.

Line 339-342. No idea what this means.

Line 353-360. Vague and unclear.

Line 420-423. Hard to follow.

Reviewer #2:

Remarks to the Author:

In this very comprehensive study, Huang et al. isolated and characterized a large panel of human monoclonal antibodies (mAbs) specific for the EV71 capsid from three children infected with this virus. To date, only mouse mAbs specific for EV71 have been studied in detail. By contrast, there is no information on the epitopes of EV71 recognized by human neutralizing mAbs, which underscores the novelty of this study. Of the 84 EV71-specific mAbs derived from plasmablasts of infected donors, 38 neutralized virus genotypes B5 and/or C4. These mAbs could be classified into 12 clonal groups based on their V(D)J rearrangements. To identify epitopes targeted by neutralizing mAbs, the authors selected escape mutants in vitro from genotype B5 and C4 viruses in the presence of selected mAbs. Remarkably, the most potent neutralizing epitopes mapped to the rims and floor of the EV71 capsid canyon, whereas 3- and 2-fold plateau epitopes were far less neutralizing. The structural and functional dichotomy between these epitopes was clear and convincing. Finally, in a mouse infection model, co-administration of EV71 with canyon floor- or canyon rim-specific mAbs was shown to prevent both motor function deficit and growth delay, suggesting that neutralizing mAbs may have therapeutic potential.

Points to address:

1. In the abstract, the statement that 191 mAbs were characterized is somewhat misleading, since only 84 of the mAbs were actually EV71-specific. The abstract should be amended accordingly.
2. Is any information available on the clinical course of EV71 infections of donors M, Y and Z? In particular, was infection less severe or of shorter duration in donor M, whose mAbs displayed substantially higher neutralization titers than mAbs from donors Y and Z?

Reviewer #3:

Remarks to the Author:

General comments.

This is an interesting manuscript in which the antibody response is assessed using samples from EV71-infected children. The study is very thorough and will be of interest to a general audience of readers of Nature Communications. While well written, the manuscript is somewhat difficult to read, partly because it is so comprehensive. The mouse experiments should also be expanded to prove antibody efficacy.

Specific comments.

1. Line 117-should be donors Y and Z instead of B and C.
2. Mut# in Table 1 requires clarification. In some instances, such as Donor M, group 6, the CDR3 amino acid sequences are identical, but the number of amino acid replacements ranges from 11 to 15. Why does the number of replacements differ if the actual sequence is the same?
3. Line 151-156-Especially for Donor Z, only a very limited number of heavy and light chains were identified (two heavy and two light chains VDJ/VJ rearrangements for donor Z). When the data may reflect a bias towards specific rearrangements, it seems possible that the results may also reflect selection during the process of in vitro analysis. This possibility should be considered.
4. Line 219, Figure 4b-How were these summary data generated? Were they averaged from individual assays or were antibodies pooled prior to use in neutralization assays?
5. Line 289-Antibodies were tested in hSCARB2 Tg mice inoculated with EV71. These mice do not develop clinical disease or weight loss but do develop motor deficits. Infection with virus-antibody mixtures showed EV71-specific antibody efficacy in preventing disease compared to mice that received virus only. A more appropriate assay for efficacy is to pretreat mice with a few concentrations of antibody and measure efficacy. Further, in addition to assessment of motor deficits, virus titers in the brain should be measured after antibody treatment.

Reviewer #1:

This paper represents a great deal of work on the part of the authors, who have generated monoclonal antibodies from plasmablasts of 3 children infected with enterovirus 71, tested the neutralizing capacity of the antibodies, and selected viral escape mutants to map the epitopes recognized by a representative subset. The results indicate that effective neutralizing antibodies recognize epitopes associated with the rims or the base of the viral canyon, and are effective against viruses of diverse genotypes; in contrast, antibodies that bind plateaus near the 2-fold and 3-fold axes neutralize weakly. The authors further test sera from a number of other infected children against the panel escape mutants, and are able to infer that some children have responses focused more on the canyon region or on the plateaus; they conclude that those whose responses are focused on the plateaus have weaker and less broadly protective neutralizing responses.

I think that, with few exceptions, the experiments are well-designed and performed, and I think the paper provides interesting new information about the human neutralizing response to EV71 infection. However, I think the paper needs extensive revision for clarity, and I am not convinced that there is a subset of children with weak neutralizing responses focused on the plateaus. I also find that some of the authors' statements about the importance of the work, and its relevance to vaccine design, etc. are unconvincing and need better justification.

- We have revised the manuscript to clarify our findings on the EV71-neutralizing antibody repertoire and modified the statements about the importance of this work and its relevance to vaccine design (Abstract, page 3; Results, pages 7, 10, 11, 13, and 14; Discussion, pages 18-20, 22, and 23; Figure 5).

Substantive issues.

1. I don't think that it is appropriate to conclude from the data presented here that, in general, children with lower neutralizing titers have antibody responses focused on the plateaus, or that children with responses focused on the plateaus have a lower neutralizing titer, as is stated in the discussion (Lines 351-353) and in the abstract. As far as I can see, the data show that in a single child with low titer, the response was focused on plateau epitopes as determined from the repertoire of monoclonal antibodies. In a group of children whose polyclonal responses were focused on the plateau epitopes, there was less activity against a heterologous virus, but the neutralizing titers against B5 were the same in both groups of children (as shown in Figure 5B). And speaking of 5B, what are the "controls" shown in the third column?

- In the previous Figure 5b, the first serum group (sera reactive to 2-fold/3-fold plateau epitopes) contains antibody repertoires that are solely reactive to plateau epitopes as well as those that are reactive to both plateau and canyon epitopes. We have revised Figure 5b and the related description in the Results section to clarify our findings with post-infection sera (page 14, lines 264-272).

- In the post-infection serum experiment, we noted that 9 out of 27 serum samples did not show a ≥ 8 -fold reduction in neutralization titers to any of the escape mutants, and these sera were presented as a control group in the third column of previous Figure 5b. We suggest that the EV71 capsid epitopes mapped by our neutralizing monoclonal antibodies may not be relevant for these nine post-infection serum samples.
- The screening of post-infection sera revealed two other pediatric patients, in addition to donors Y and Z, whose sera were predominantly reactive to plateau epitopes (Fig. 5b). The average serum neutralization titer in patients with antibodies that were predominantly reactive to plateau epitopes was significantly lower than the titer of sera that were reactive to both canyon and plateau epitopes or those that were predominantly reactive to canyon epitopes (Fig. 5b). However, we also agree that some children with low neutralization titers may not always have antibody responses focused on the plateau epitopes. In addition, the EV71 capsid epitopes mapped by our neutralizing mAbs may not be relevant in some post-infection sera, and the neutralization epitopes targeted by these sera are unknown. We have revised the related descriptions in the Abstract and Discussion sections to clarify our findings (page 3, lines 31-34; page 13, lines 252-258; page 14, lines 264-272).

2. Figure 2B. The effective dilutions of monoclonal antibody supernatants can't be used to compare antibodies unless we know that the antibody concentrations are the same. Reading the methods describing the binding curves I'm not sure what was measured here. Counting the number of infected cells has nothing to do with the binding avidity of the antibody. I think there was a known number of infected cells (determined how?) and this was counted as 100%. So the percentage is the number of infected cells detected (as a percentage of the total infected cells) rather than the number of infected cells (as a percentage of the total cells).

I don't understand the statement (lines 166-168) discussing 2B. To me, "while" means "at the same time." And I see no reason why the number of infected cells detected with a particular antibody should relate to its neutralizing efficacy.

- We have revised the figure and presented the effective concentrations of the EV71 mAb supernatants in the revised manuscript (Table 1 and Fig. 1) (pages 40 and 43).
- We have further clarified the flow cytometry-based binding assay method in the revised manuscript (Methods, page 26, lines 524-531). The infected cells were detected with a mouse anti-EV71 3C (a non-structural protein) antibody and probed with a fluorescein-conjugated secondary antibody. The percentage of infected cells was derived from an analysis of 10,000 gated events of EV71-infected (3C-positive) cells, and is shown as the percentage of EV71-infected cells that bound detectable human anti-EV71 antibodies.
- We agree that the percentage of infected cells bound by the human anti-EV71 monoclonal antibody in the flow cytometry assay should not be related to the neutralizing efficacy of the antibody. Therefore, we have revised the statement in the Results section of the manuscript (page 10, lines 167-183).

3. Donor Z had a low neutralizing titer, and monoclonals generated from his cells had weak activity against the genotype B5 virus used for mapping epitopes. As far as I know, the various genogroups of EV71 are determined on the basis of sequence rather than on serological properties. Do we know that all B5 viruses are serologically the same? Is there any possibility that this child was infected with a B5 variant that was serologically distinct from the virus isolate tested in the lab? Was the child's virus sequenced in its entirety? Maybe he had a terrific response against the virus with which he was really infected. Can we exclude the possibility that it was a recombinant in which the EV71 neutralizing epitopes were replaced by sequences from another virus. This may seem far-fetched, but the authors are placing a lot of weight on the results from this single child.

- The genotypes of EV71 are most commonly determined based on phylogenetic and sequence analyses of capsid VP1 genome (refs. 2-6, 8, 9, 11). The capsid VP4 and VP2 sequences are also used to classify human enteroviruses into genotypes. The typing of human enterovirus A species (i.e. EV71, Cox A16) by VP4 or VP2 shows 100% concordance with typing by VP1 (Perera et al., 2010). The current genotyping system demonstrates the range of EV71 genetic diversity, and the emergence and re-emergence of genotype is tightly associated with the outbreaks in Europe and the Asia-Pacific region (refs. 1-7).
- We sequenced the entire P1 region (encoding structural proteins VP1–4) of the clinical isolates from the three mAb donors. The clinical isolates from donors M (12-Taoyuan16), Y (12-Taoyuan17), and Z (12-Taoyuan34) are highly related to each other (P1 nucleotide identity 98.4–99.1% and P1 amino acid identity 99.1–100%) and are phylogenetically clustered into the B5 genotype. We also analyzed the complete genome of the clinical isolate from donor Z (12-Taoyuan34) and other reference isolates collected in 2012. The complete genome sequence of 12-Taoyuan34 is at least 97.7% identical and the deduced amino acid sequence is at least 99.5% identical to all other 2012 EV71 reference isolates. These 2012 EV71 reference isolates, including 12-M538, 12-M1089, 12-M202, 12-M988, and 12-M617, were previously published (ref. 5), and no obvious recombination between the 2012 EV71 isolates and other enteroviruses was detected. We have provided additional information about our sequence analysis of EV71 clinical isolates in the revised manuscript (Supplementary Fig. 1). The sequences of the 12-Taoyuan16, 12-Taoyuan17, 12-Taoyuan34, 12-1616, 12-96059, 14-51389, 15-921, 99-3351, 07-72043, and 98-2086 EV71 viruses have been submitted to GenBank.
- The antigenic variation among EV71 genotypes has been studied using a panel of murine mAbs, and the results suggest that the current genotyping of EV71 does not perfectly reflect their antigenicity (ref. 22). However, there are limited data about the antigenic properties of EV71 capsid with human antibodies; the available data are from analyses of human sera collected from vaccinated and infected individuals (refs. 6, 15-18). For the genotype B5 EV71 viruses (12-Taoyuan16, 12-Taoyuan17, and 12-Taoyuan34) isolated from the mAb donors, the three viruses are recognized by individual donor serum at titers within a 2-fold range, suggesting that these viruses are

antigenically similar.

4. Do we know that plasmablasts isolated at day 9 (or serum taken at day 11) reflect the breadth of the antibody response that will ultimately determine protection from reinfection? Does the response continue to mature after that, with selection for higher-affinity clones?

- In a previous study, we found that the circulating EV71-specific plasmablast response in infected children most frequently peaked 4–7 days after illness onset, followed 1–3 days and 8–11 days after onset (average frequency, 1540 ± 979 , 343 ± 410 , 80 ± 98 cells per million PBMCs, respectively) (ref. 15). The average frequency of plasmablasts on days 8–11 was 10-fold lower than that on days 4–7 after onset. In most cases, the plasmablast response disappeared two weeks after infection. Therefore, to isolate and characterize the antibody repertoire that is generated in response to natural EV71 infection, we utilized circulating plasmablasts collected on ‘day 7’ after illness onset (Introduction, page 5, lines 82-84; Results, page 8, lines 119-122; Discussion, page 19, lines 374-378). We found that the EV71-specific plasmablast response is characterized by high levels of somatic mutations and significant clonal expansions (Fig. 1, Table 1 and Supplementary Table 1). The neutralization breadth and potency of plasmablast-derived mAbs in our donors were associated with the convalescent serological response on days 9–11 (Fig. 1 and Fig. 5).
- Plasmablasts are activated antibody-secreting B cells that are transiently present in the peripheral blood after vaccination or infection (refs. 15, 19, 38, 41, 42, 51), and they survive in the plasma cell niches of mucosal tissue, lymphoid organs, and bone marrow for months to years (Odendahl et al., 2005; Mei et al., 2009). It is generally believed that antibody-secreting and memory B cells constitute the primary cellular components of the T cell-dependent antibody response to a variety of viral pathogens (refs. 33, 34, 37, 38, 40, 41, 42, 52).
- There is still considerable debate regarding the mechanisms by which long-term immunity is maintained by antibody-secreting and memory B cells. A longitudinal analysis of antibody titers specific for six viral antigens in 45 adults for a period of up to 26 years has shown that the B-cell memory is long-lived, and the correlation between peripheral memory B-cell numbers and antibody levels is significant for measles, mumps, and rubella viral antigens (Amanna et al., 2007). Increasing evidence shows that the plasmablast response might serve as a surrogate marker for the antibody response to viral antigens. Following influenza vaccination, the magnitude of the virus-specific plasmablast response significantly correlates with the serological titer in humans (Halliley et al., 2010; Wrammert et al., 2012; Wu et al., 2011). Antibody-mediated immunity plays a critical role in mediating protection against EV71 illness. Nevertheless, we acknowledge that a longitudinal and detailed analysis of the dynamics of the EV71-specific B cell repertoire and the correlation between the magnitude of B cell clonal expansion and antibody response during the acute and convalescent stages of infection is still lacking at present.

- We have added related details and described the limitation of our study in the revised manuscript (Results, page 8, lines 119-122; Discussion, page 19, lines 365-380; Discussion, page 20, lines 381-390).

5. I don't think the experiments with antibodies added before and after attachment are easy to interpret (and I'm not sure what the authors really make of them). All the antibodies inhibit in both situations, and virtually all of them show reduced activity after virus has already bound. The weaker antibodies do worse in both cases. I don't think that these experiments really can be used to support the idea that some antibodies directly block interaction with receptors and others do not. Antibodies could cause virus clumping, they could stabilize viruses against uncoating, they could trigger premature uncoating. More evidence would be needed to distinguish among the many possibilities

- We agree that the details of the neutralization mechanisms of the mAbs require further investigation; however, this is beyond the scope of our present study. We have added more details to the revised manuscript and described the limitations of our study (Results, page 14, lines 281 and 282; Discussion, page 21, lines 420-422; Discussion, page 22, lines 439-441).
- We are currently collaborating with Prof. David Stuart at the University of Oxford to study the Fab-virus complex and the interaction between virus and antibody at the structural level, which will be helpful for determining the neutralization mechanisms.

6. In one place it's said that the 2 most potent antibodies were effective at 24 and 98 ng/ml, but something slightly different is shown in 2C. If the specific numbers aren't important, why be so specific?

- In the previous manuscript, we stated that 'Antibodies 16-3-10B and 16-2-8C were two most potent clones here, which retained the neutralization activity at 24 and 98 ng/mL', which was based on the data for the neutralizing activities of purified representative mAbs against EV71 genotype C4 11-96023 and B5 12-96015 viruses, as shown in previous Figure 2b. To explore the neutralization breadth of representative mAbs, they were tested against EV71 clinical strains isolated in 1998–2016. We agree that the potent and broadly neutralizing mAbs exhibit varying degrees of neutralization potency against the tested viruses (previous Fig. 2c, Table 2 in the revised manuscript). Therefore, we have revised the description of antibody potency in the manuscript (Results, page 10, lines 167-183).

7. In the mapping of the epitopes, you must have sequenced the genomes of escape mutants, but sequencing is mentioned nowhere.

- Yes, we have sequenced the complete P1 genome region (VP1–4) of each escape mutant. We have further clarified the method in the revised manuscript (Methods, page 28, lines 574 and 575).

8. Paragraph beginning 203 needs more information and rewording to be meaningful.

- We have revised the description of the site-directed mutagenesis data in the manuscript (Results, page 11, lines 202-210).

9. In Figure 3A, are the different residues involved in escape from each particular antibody (for example VP1 M229 and VP2 T232, or VP3 E81 and VP2 D225) located close to each other on the capsid surface?

- Figure 3a shows that one neutralizing antibody might select for different substitutions at the same residue or more than one residue replaced in the escape mutants; however, such residues tend to cluster, and they are near either in the sequence or adjacent to each other in three-dimensional space (Fig. 3b). Taken together, this suggests that the footprint of one neutralizing antibody on the viral capsid may be slightly different among EV71 strains but the neutralizing epitope engaged by the antibody appears to be unique and highly specific. We have designated the residues involved in the escape mutants for each antibody in Figure 3b of the revised manuscript (page 45).

Issues of presentation:

The paper needs careful review by a professional editor to correct numerous small errors in grammar and diction— especially the use or omission of articles and the choice of plural vs. singular. The tenses should be brought into accord with the general convention for scientific papers, in which the present tense is used to describe generally accepted facts, published results, and major conclusions, whereas the past tense is used to describe the results of specific experiments described in the results section. In many cases these glitches are just an annoyance to the reader, but in some cases they are truly confusing. I list some of these below, but there are many more.

- We have corrected all errors in grammar and diction in the revised manuscript. We have carefully proofread the paper to ensure that we used the proper articles and correct tenses.

In a number of places the text contained in the figures is very small and difficult to read. For example, the table in 2C could be split off as a table and made larger, the sequences in 3A and the labeling of residues in 3B are virtually illegible. In Fig 5A it would be helpful to indicate the location of specific residues, as they are marked or colored elsewhere; in the key to the figure "fold decrease..." is tiny, and the dark brown color hides the numbers (?8 and 16?).

- We have revised Figure 2, and we have moved the data in previous Figure 2c to Table 2 of the revised manuscript. We have also revised Figures 3 and 5 in the manuscript (pages 45 and 49).

Examples of presentation problems:

The title is uninformative. "Phenomena" makes it sound like the paper is about random observations rather than a scientific inquiry.

- We have revised the title in the manuscript (page 1).

Abstract: I don't think there is such a thing as an "endemic." What is a "multi-genotype" virus? It's not obvious to what implications the work has for evaluation of vaccines or antibody therapeutics. Vaccines will be tested empirically for generation of neutralizing responses and protective efficacy.

- We have revised the text in the Abstract section of the manuscript (page 3).

Introduction: Line 44, outbreaks "are" associated with. Line 58, "alternative" to what? Line 68 "mouse" or "murine." Line 92 "unravel the antigenic landscape" Huh?

- We have revised the text in the Introduction section of the manuscript (page 4, lines 40, 54, and 55; page 5, line 64; page 6, lines 86-88).

Results: Line 102 referred to AS donors... Average age is silly, given that there are only 3 subjects. Line 109. Paired sera.

- We have revised the text in the Results section of the manuscript (page 7, lines 99-102).

Line 114. Each donor developed "distinct" antibody response. In what way?

- We have revised the text in the Results section of the manuscript (page 7, lines 111-113).

Line 117. small-titer increase? ?small increase in titer?

- We have revised the text in the Results section of the manuscript (page 7, lines 113-116).

Who are donors B and C? And why would patients exposed to one virus mount a response to another virus. Rephrase— ? the antibody produced in response to infection neutralized B5 but failed to neutralize...

- We have revised the text in the Results section of the manuscript (page 7, lines 116-118).

Line 134. We first observed an efficient development of neutralizing clones in 3 donors.. I don't understand the use of "first" and I don't understand the repeated use mention 3 donors. Were there other donors in whom you didn't observe this? Do you mean 'all 3 donors?'"

- We have revised the text in the Results section of the manuscript (page 8, lines 133-141).

Line 149, 152. repertoires WERE;

- We have revised the text in the Results section of the manuscript (page 9, lines 150-154).

Lines 159 and 160. I have no idea what this means. Please explain

- We have revised the text in the Results section of the manuscript (page 9, lines 150-159).

Line 163. map the releant epitopeS.

- We have revised the text in the Results section of the manuscript (page 9, lines 161-164).

Line 174. broadly reactive

- We have revised the text in the Results section of the manuscript (page 10, line 170).

Line 181. "Confirmed"? Is this the confirmation of someone elses's result? Of a result you discussed earlier in the paper?

- We have revised the text in the Results section of in the manuscript (page 10, lines 167-181).

Discussion:

Line 308. "responsible." This is a supposition. What you know is they were associated.

- We have revised the text in the Discussion section of the manuscript (page 17, lines 311-313).

Line 311: 'focused on the overlapping regions on the capsid canyon with potent activities." Makes no sense as written.

- We have revised the text in the Discussion section of the manuscript (page 17, lines 313-317).

Line 339-342. No idea what this means.

- We have revised the text in the Discussion section of the manuscript (page 18, lines 338-344).

Line 353-360. Vague and unclear.

- We have revised the text in the Discussion section of the manuscript (page 18, lines 350-355; page 19, lines 356-364).

Line 420-423. Hard to follow.

- We have revised the text in the Discussion section of the manuscript (page 22, lines 430-441).

Reviewer #2:

In this very comprehensive study, Huang et al. isolated and characterized a large panel of human monoclonal antibodies (mAbs) specific for the EV71 capsid from three children infected with this virus. To date, only mouse mAbs specific for EV71 have been studied in detail. By contrast, there is no information on the epitopes of EV71 recognized by human neutralizing mAbs, which underscores the novelty of this study. Of the 84 EV71-specific mAbs derived from plasmablasts of infected donors, 38 neutralized virus genotypes B5 and/or C4. These mAbs could be classified into 12 clonal groups based on their V(D)J rearrangements. To identify epitopes targeted by neutralizing mAbs, the authors selected escape mutants in vitro from genotype B5 and C4 viruses in the presence of selected mAbs. Remarkably, the most potent neutralizing epitopes mapped to the rims and floor of the EV71 capsid canyon, whereas 3- and 2-fold plateau epitopes were far less neutralizing. The structural and functional dichotomy between these epitopes was clear and convincing. Finally, in a mouse infection model, co-administration of EV71 with canyon floor- or canyon rim-specific mAbs was shown to prevent both motor function deficit and growth delay, suggesting that neutralizing mAbs may have therapeutic potential.

Points to address:

1. In the abstract, the statement that 191 mAbs were characterized is somewhat misleading, since only 84 of the mAbs were actually EV71-specific. The abstract should be amended accordingly.

- We have revised the Abstract of the manuscript (page 3, lines 28-31).

2. Is any information available on the clinical course of EV71 infections of donors M, Y and Z? In particular, was infection less severe or of shorter duration in donor M, whose mAbs displayed substantially higher neutralization titers than mAbs from donors Y and Z?

- All three donors were diagnosed with hand-foot-and-mouth disease, and had fever that lasted for 4 days in the acute phase of EV71 infection, but none developed neurological symptoms. All three donors completely recovered within two weeks. We have provided additional information about the clinical course of infection in the donors to the Results section of the revised manuscript (page 7, lines 99-103).

Reviewer #3:

General comments.

This is an interesting manuscript in which the antibody response is assessed using samples from EV71-infected children. The study is very thorough and will be of interest to a general audience of readers of Nature Communications. While well written, the manuscript is somewhat difficult to read, partly because it is so comprehensive. The mouse experiments should also be expanded to prove antibody efficacy.

- We have further clarified our methods, provided additional information in the Results and Discussion sections and corrected errors in grammar and diction in the revised manuscript.

Specific comments.

1. Line 117-should be donors Y and Z instead of B and C.

- We have revised the text in the Results section of the manuscript (page 7, lines 116-118).

2. Mut# in Table 1 requires clarification. In some instances, such as Donor M, group 6, the CDR3 amino acid sequences are identical, but the number of amino acid replacements ranges from 11 to 15. Why does the number of replacements differ if the actual sequence is the same?

- We have provided additional information about the numbers of nucleotide mutations and amino acid replacements in the footnote of Table 1 in the revised manuscript (page 40, lines 797-801). The data shown in column 6 of the table are the numbers of nucleotide mutations and amino acid replacements in the 'variable domains' of the mAbs. The variable domain consists of framework regions (FR1, FR2, FR3, and FR4) and complementarity determining regions (CDR1, CDR2, and CDR3). The heavy chain and light chain variable domain sequences are aligned with germline gene segments using the international ImMunoGeneTics (IMGT) alignment tools to determine the individual gene segments employed by the VDJ and VJ rearrangements and the number of mutations.
- In column 5 of the Table 1, we presented the amino acid sequence of the heavy chain VDJ junction, which consists of one conserved Cys (the last residue of FR3), the CDR3, and one conserved Trp (the first residue of FR4), as determined by the IMGT/JunctionAnalysis. The plasmablast immunoglobulin sequences that shared identical VDJ and VJ rearrangements and similar junction regions were considered to be clonally related. However, each mAb was derived from an individual plasmablast and encoded by a unique sequence.
- For example, 10 neutralizing antibodies derived from the circulating plasmablasts of donor M share identical heavy chain variable domain V_H 4-39*01/D_H 6-19*01/J_H 6*02 and light chain variable domain V_λ 1-44*01/J_λ 3*02 rearrangements and are grouped into one closely related clonal group

(Table 1). The pairwise nucleotide identities for the heavy chain variable domains (378 nt) were 96.0–99.7%, and those for the light chain variable domains (330 nt) were 98.2–99.4%. The pairwise amino acid identities for the heavy chain variable domains (126 aa) were 94.4–99.2%, and those for the light chain variable domains (110 aa) were 97.3–99.1%. We analyzed the heavy chain variable domain sequences using the IMGT alignment tool, and the results showed that there are 1–3 amino acid replacements in FR1 (25 aa), 0–4 replacements in CDR1 (10 aa), 1–2 replacements in FR2 (17 aa), 2 replacements in CDR2 (7 aa), 6–7 replacements in FR3 (38 aa), 0 replacement in CDR3 (18 aa), and 0 replacement in FR4 (11 aa) of the heavy chain variable domain of group 6 antibodies. Taken together, the total number of amino acid replacements in the heavy chain variable domains ranges from 11–15 for group 6 antibodies.

- The sequences of the variable domains of all representative antibodies have been deposited in GenBank (page 10, lines 181-183).

3. Line 151-156-Especially for Donor Z, only a very limited number of heavy and light chains were identified (two heavy and two light chains VDJ/VJ rearrangements for donor Z). When the data may reflect a bias towards specific rearrangements, it seems possible that the results may also reflect selection during the process of in vitro analysis. This possibility should be considered.

- We agree that we have isolated a limited number of plasmablasts and acknowledge that this study has several limitations. First, the antigen-specific plasmablasts that we isolated to make mAbs were elicited during the acute infection with a particular strain of EV71, and therefore may be enriched for an antibody repertoire specific to recent antigen exposure. Second, IgM and IgA plasmablasts were not used to generate mAbs. Upon acute EV71 infection, virus-specific IgG plasmablast responses are predominant among children >3 years of age; however, this IgG response is sometimes accompanied by a nearly equivalent IgM plasmablast response (ref. 15). Therefore, the clonality and function of EV71-specific IgM and IgA antibodies were not determined in our donors. Third, the plasmablast-derived antibody clones may not represent the complete repertoire of EV71-specific B cells, as the memory B cell repertoire was not analyzed in the study (Discussion, page 19, lines 365-380; Discussion, page 20, lines 381-390).
- Nevertheless, our clonal analysis of plasmablast-derived antibodies from donor Z revealed an abundant subset of IgG clones available for recognizing EV71 capsid. The 25 EV71 capsid-specific antibodies from donor Z could be classified into 12 clonal groups based on the VDJ/VJ rearrangements and junction regions of the variable domain sequences (Table 1 and Supplementary Table 1). Of the 25 EV71-specific antibodies, 11 neutralized viruses and were clustered into one clonal group (V_H 7-4-1*02/ D_H 1-7*01/ J_H 6*02 and V_K 1-39*01 or 1D-39*01/ J_K 4*01; Table 1). The reason for the clonal expansion of a particular plasmablast lineage in response to acute EV71 infection in donor Z is unclear; however, possible mechanisms have been discussed in the manuscript (Discussion, page 17, lines 317-324). Bias toward particular gene rearrangements in the variable domain has also been observed

in influenza HA stem- (V_H 1-69), HIV V3- (V_H 5-51), and rotavirus-specific (V_H 1-46) antibodies derived from donors after infection (refs. 33-35).

4. Line 219, Figure 4b-How were these summary data generated? Were they averaged from individual assays or were antibodies pooled prior to use in neutralization assays?

- Figure 4b shows box and whisker plots of the neutralizing activities ($\mu\text{g/mL}$) of canyon mAbs (grey dots box) and plateau mAbs (empty box). The values were averaged from the data for each representative mAb. The neutralizing activity of each mAb is defined as the lowest concentration that completely inhibited CPE formation. Each antibody was assayed in triplicate for each virus, and the assay was carried out twice with equivalent results. We have provided additional information to explain the data presented in the Figure 4b in the revised manuscript (page 48, lines 860-866).

5. Line 289-Antibodies were tested in hSCARB2 Tg mice inoculated with EV71. These mice do not develop clinical disease or weight loss but do develop motor deficits. Infection with virus-antibody mixtures showed EV71-specific antibody efficacy in preventing disease compared to mice that received virus only. A more appropriate assay for efficacy is to pretreat mice with a few concentrations of antibody and measure efficacy. Further, in addition to assessment of motor deficits, virus titers in the brain should be measured after antibody treatment.

- To establish the hSCARB2-transgenic mouse challenge model with a non-mouse-adapted EV71 clinical strain (C4 11-96023 or B5 12-96015; GenBank accession numbers KF154296.1 and KF154355.1, respectively), different age groups of mice, several different routes and dosages of infection were examined (ref. 39). The results showed that intracerebral challenge with 1×10^6 or 4×10^6 TCID₅₀ of EV71 results in significant motor deficit and growth retardation at 5–9 days post infection. The motor deficit scores increased dose-dependently. For the in vivo protection experiments with mAbs, a challenge dose of 4×10^6 TCID₅₀ of EV71 was utilized, and the antibody dose was estimated based on its in vitro potency in the present study. This virus-antibody mixture method has also been used to examine the in vivo protective efficacy of antibodies against intracerebral infection by enteroviruses (Cao et al., J. Clin. Virol. 2011; Mao et al., J. Virol. 2012; Jiang et al., Vaccine 2015). In these studies, a high antibody concentration is often required to confer >80% protection against severe illness and mortality, although the infective dose might vary.
- We agree that further investigations of different doses, time points, and routes of administration of these neutralizing mAbs in small and large animal models are important for determining the prophylactic and therapeutic efficacies of these antibodies; however, these are beyond the scope of the current study. In addition, proper setup of the animal challenge models with clinical EV71 strains that mimic all the clinical manifestations in humans would be important for examining the efficacy of antibodies, anti-viral molecules, and vaccines. We have revised our interpretation and significance of the animal data in the Abstract and Discussion sections of the manuscript (page 3, lines 28-36; page 23, lines 456-562).

- We harvested brain tissues from mice at the end of the experiment (at 14 days post-infection), and the tissue samples were homogenized in sterile phosphate-buffered saline, disrupted by three freeze-thaw cycles, and centrifuged. The viral titer of the supernatants was examined by the TCID₅₀ assay. We found that the virus was barely detectable in the brain tissues. No CPE was detected in cell culture replicates infected with the initial dilution (10^{-1}) of most supernatants from both the virus-buffer and virus-mAb groups. There are two possible explanations for the difficulty in determining the viral titer in our samples. First, we harvested samples at 14 days post-infection, a time point when neurological symptoms have subsided and the majority of symptomatic mice have recovered from the motor deficit caused by the intracerebral challenge (Fig. 7). The non-mouse-adapted EV71 clinical strain may replicate initially but inconsistently in brain tissue, which would also explain why the motor impairment peaks at day 5–7 and gradually improves after 9 days post-infection in the challenged mice. Therefore, the viral level in brain tissue at 14 days post-infection may have been too low to be detected by the TCID₅₀ assay. Second, in the challenge study, a small volume of the virus preparation (5 μ L of virus + 5 μ L of buffer or mAb) was injected into the striatum of brain. Since whole brain tissue was harvested and tissue supernatant was prepared, the infectious viruses would have been greatly diluted. A more sensitive method, i.e. real-time quantitative reverse transcriptase PCR (qRT-PCR), could be used to determine the viral titer. However, the above method was not performed in the study because there was not enough sample left after repeated TCID₅₀ experiments.

Reviewers' Comments:

Reviewer #1:

Remarks to the Author:

Huang ... Lin

Nature Communications June 2017 (REV1)

Epitope-associated and specificity-focused features of EV71-neutralizing antibody...

The authors have responded to all of my major comments and questions and this revised manuscript is much clearer and easier to read. The results are convincing and provide novel insights into the human immune response to this clinically important virus.

I have only a few minor comments on presentation:

Lines 85-86. Unclear. ? some sera from children infected with genotype B viruses show low cross-neutralization activity against genotype C?

Line 109. no titer change AGAINST CVA2 was detected...

Lines 336-337. Clarify that the children were infected with B5, and failed to develop antibodies cross-protective against C4?

Jeff Bergelson

Reviewer #2:

Remarks to the Author:

The authors have responded satisfactorily to the previous critiques.

Reviewer #3:

Remarks to the Author:

The authors have responded very well to previous comments in all instances with one exception. The mouse studies are not improved. However, animal studies may not be necessary because the antibody analyses are so complete. Improvement of the animal studies would require examining the effects of antibody treatment on virus replication at earlier times after infection, not when the infection has mostly resolved, and delivering antibody prior to infection rather than by mixing with the virus. Some of the studies referenced by the authors use this approach, so it is feasible. Since these additional mouse experiments will take time and are likely to be costly, and the in vitro analyses of the human antibody response are well done and thorough, consideration should be given to deleting them entirely from the manuscript.

Reviewer #1:

The authors have responded to all of my major comments and questions and this revised manuscript is much clearer and easier to read. The results are convincing and provide novel insights into the human immune response to this clinically important virus.

I have only a few minor comments on presentation:

Lines 85-86. Unclear. ? some sera from children infected with genotype B viruses show low cross-neutralization activity against genotype C?

- We have revised the text in the Introduction section of the manuscript (page 6, lines 86-87).

Line 109. no titer change AGAINST CVA2 was detected...

- We have revised the text in the Results section of the manuscript (page 7, line 110).

Lines 336-337. Clarify that the children were infected with B5, and failed to develop antibodies cross-protective against C4?

- We have revised the text in the Discussion section of the manuscript (page 17, lines 318-320).

Reviewer #3:

The authors have responded very well to previous comments in all instances with one exception. The mouse studies are not improved. However, animal studies may not be necessary because the antibody analyses are so complete. Improvement of the animal studies would require examining the effects of antibody treatment on virus replication at earlier times after infection, not when the infection has mostly resolved, and delivering antibody prior to infection rather than by mixing with the virus. Some of the studies referenced by the authors use this approach, so it is feasible. Since these additional mouse experiments will take time and are likely to be costly, and the in vitro analyses of the human antibody response are well done and thorough, consideration should be given to deleting them entirely from the manuscript.

- We have deleted the text about the mouse study in the manuscript (page 6, lines 90-91; page 15; page 22, lines 437-438; page 29).